# Many Miles to Paris: A Sectoral Innovation System Analysis of the Transport Sector in Norway and Canada in Light of the Paris Agreement

**Konstantinos Koasidis** [1], **Anastasios Karamaneas** [1], **Alexandros Nikas** [1,*], **Hera Neofytou** [1], **Erlend A. T. Hermansen** [2], **Kathleen Vaillancourt** [3] and **Haris Doukas** [1]

[1] Decision Support Systems Laboratory, School of Electrical and Computer Engineering, National Technical University of Athens, Iroon Politechniou 9, 157 80 Athens, Greece; kkoasidis@epu.ntua.gr (K.K.); akaramaneas@epu.ntua.gr (A.K.); hneofytou@epu.ntua.gr (H.N.); h_doukas@epu.ntua.gr (H.D.)

[2] CICERO Center for International Climate and Environmental Research, P.O. Box 1129, Blindern, NO-0318 Oslo, Norway; erlend.hermansen@cicero.oslo.no

[3] ESMIA Consultants, Blainville, QC J7B 6B4, Canada; kathleen@esmia.ca

[*] Correspondence: anikas@epu.ntua.gr; Tel.: +30-210-772-3612

**Abstract:** Transport is associated with high amounts of energy consumed and greenhouse gases emitted. Most transport means operate using fossil fuels, creating the urgent need for a rapid transformation of the sector. In this research, we examine the transport systems of Norway and Canada, two countries with similar shares of greenhouse gas emissions from transport and powerful oil industries operating within their boundaries. Our socio-technical analysis, based on the Sectoral Innovation Systems approach, attempts to identify the elements enabling Norway to become one of the leaders in the diffusion of electric vehicles, as well as the differences pacing down progress in Canada. By utilising the System Failure framework to compare the two systems, bottlenecks hindering the decarbonisation of the two transport systems are identified. Results indicate that the effectiveness of Norway's policy is exaggerated and has led to recent spillover effects towards green shipping. The activity of oil companies, regional and federal legislative disputes in Canada and the lack of sincere efforts from system actors to address challenges lead to non-drastic greenhouse gas emission reductions, despite significant policy efforts from both countries. Insights into the effectiveness of previously implemented policies and the evolution of the two sectoral systems can help draw lessons towards sustainable transport.

**Keywords:** Norway; Canada; electric mobility; transportation; socio-technical transitions; climate policy; sectoral innovation systems; system failure framework; systems of innovation

## 1. Introduction

The Paris Agreement set the target of limiting the rise of global temperature well below 2 °C. During the last decades, power generation has been the biggest worldwide pollutant, followed by transportation [1]. The latter plays a critical role in people's daily life, satisfying personal as well as commercial needs. National economies heavily depend on the development of this sector, but its environmental impact dictates the need for drastic decarbonisation. Most transport means operate using fossil fuels and are responsible for 57% of the global oil demand [2]. The sector accounts for 23% of the world's final energy use and 14% of the global anthropogenic greenhouse gas (GHG) emissions [3]. Therefore, the transformation of transport worldwide, in line with efforts towards reaching a global low-carbon economy, should be intensified.

Both passenger and freight transport are relevant for examining the diffusion of innovation in transportation, since they contribute almost equally to the sector's final consumption [4]. Road, rail, water and aviation are the major motorised transport modes. Apart from energy efficiency improvements, transport decarbonisation entails the penetration of low-carbon technologies, such as electrification, as well as the exploitation of less conventional fuels, such as biofuels and hydrogen. In particular, electric vehicles (EVs) are broken down into three categories: battery electric vehicles (BEVs), depending exclusively on electric energy; plug-in hybrid electric vehicles (PHEV), which operate similarly to ordinary hybrid cars, with the only difference being they can also be charged; and fuel cell electric vehicles (FCEV), which use hydrogen to produce the electricity required to meet their energy demands—although, considered as EVs, they mainly depend on hydrogen [5].

Accretive technological improvements alone are not sufficient to cope with sustainability challenges [6]. Socio-technical systems require deep changes in order for these challenges to be addressed. Aiming to understand the systemic change, system thinking and design are important factors [7]. Studies on innovation systems enable the understanding of the dynamics of socio-technical change through the connections between actors and networks, institutions, knowledge and technologies [8,9].

The scope of this study is to map the transport sector of Norway and Canada from a low-carbon transition perspective, by examining the structure and functions of their innovation systems and assessing potential barriers that could hinder the use and diffusion of sustainable technologies. Drawing from the examination and comparison of the systems, our objective is to gain insights into the effectiveness of implemented policies, inform on emerging pleas from within the systems and provide lessons to policymakers regarding the next steps of a transition to sustainable transport. The two countries were selected based on their profiles, which are similar in terms of GHG emissions, since the transportation sector has historically been a major emitter with similar shares in both systems, accounting for almost 24% of each country's total emissions. Oil and gas extraction is a common dominant emitter [10], while the potential or penetration of renewables is also similar. The two countries are different in terms of innovation diffusion: although they both feature high shares of transport emissions as well as a high contribution of renewable energy sources (RES) in the national energy mix [1], their EV shares in the transport mix differ significantly.

This difference enables the knowledge transfer between the two systems, leading to effective policy recommendations. Norway's initiatives can provide valuable lessons to Canada. However, the exchange of knowledge is not only from Norway to Canada, since the progress achieved by Norway is shown to be limited and does not place the country in a fundamentally better condition. From that perspective, Canada can also provide lessons, acting as a benchmark for Norway to properly evaluate policies, without exaggerating their effectiveness. At the same time, both countries are locked into their domestic oil reserves, but in a different manner: although Norway holds significantly less oil reserves, a share (usually less than 3%) of the surplus of its state-owned oil fund (the world's largest of its kind) is used to balance the national budget.

To support our analysis, we build on the Sectoral Innovation Systems (SIS) framework [11], which enables us to examine the multidimensional dynamics [9] of the transport sector, aiming to identify the role of different actors in driving the transition towards carbon lock-out. This comparison is further elaborated by means of the System Failures (SF) framework [12], which investigates potential barriers to the diffusion of a specific technology—in this case low-carbon transportation methods, which are essential for climate change mitigation. In a review of sustainable transport research, Zhao et al. [13] showed that the discussion surrounding transport policy places significant emphasis on sustainable transport indicators, which however failed to effectively influence policy. Instead, the combination of the two qualitative frameworks selected in our study allows us to delve into market and public responses, which are decisive for assessing the effectiveness of policies without exaggerations [14].

## 2. Background

Norway is placed 61st in the list of global $CO_2$ emitters, whereas Canada is the 11th biggest emitter, accounting for almost 1.5% of the global emissions [15]. Currently, the major energy sources in both countries are oil products and electricity, while in Canada natural gas also holds a significant share (Figure 1). A distinct characteristic of Norway is that 95% of the electricity produced comes from hydropower, while the respective share in Canada is about 60% (Figure 2) [16]. However, large inequalities in the distribution of Canada's resources create significant differences among the provinces [17], as is the case with the abundant hydropower potential in the region of Quebec [18]. This could be a factor for enabling the transition to EVs, especially in certain regions with an increased share of renewable energy production. On the other hand, Canada is among the top 5 oil producers and top 10 oil consumers, owning a great number of domestic oil reserves, accounting for 10% of the world's reserves [19]. This might hinder the energy transition of Canada's transport sector, since domestic reserves ensure reliability and security of supply. Norway owns a number of domestic oil reserves as well, but they are 20 times lower than Canada's, and two thirds of its oil production is exported [20,21].

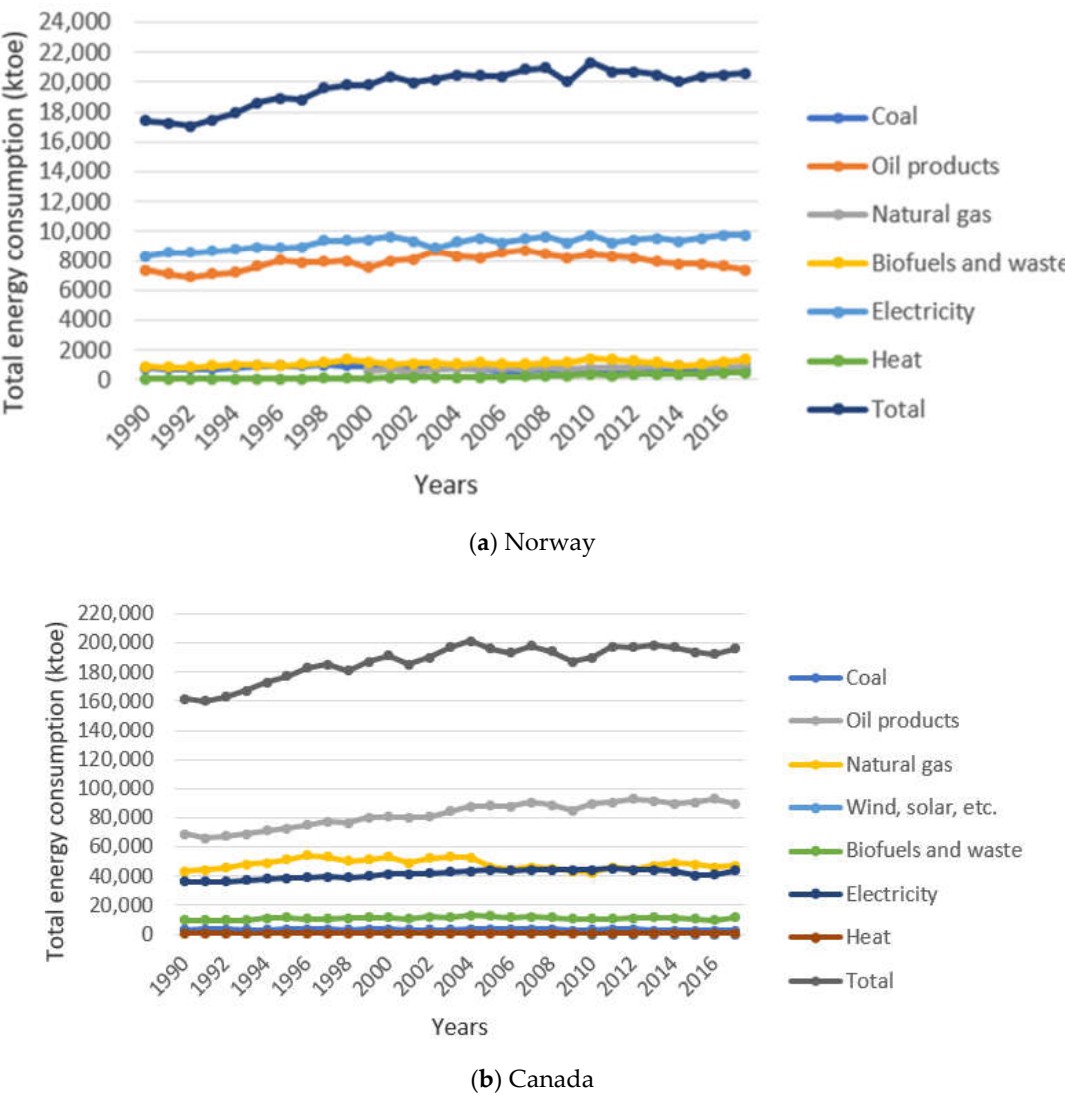

(**a**) Norway

(**b**) Canada

**Figure 1.** Total energy consumption per energy source in (**a**) Norway, and (**b**) Canada. Source: [16], own elaboration.

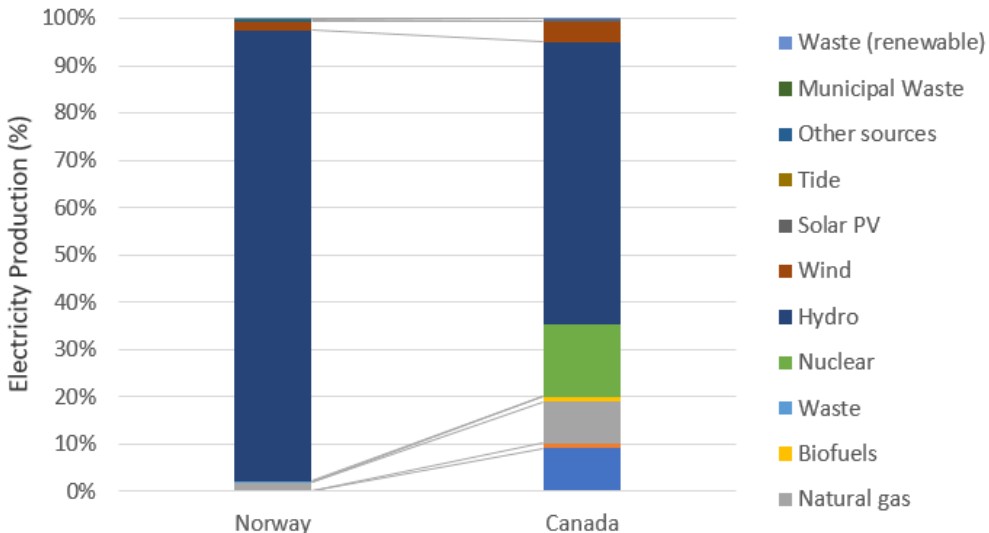

**Figure 2.** Comparative electricity production per energy source in Norway and Canada. Source: [16], own elaboration.

The transport sector contributes to the final energy consumption with 22% and 31% in Norway and Canada, respectively, and is dominated by the use of oil products, as shown in Figure 3. It should also be mentioned that the oil contribution in Norway's transport sector has dropped since 2011, with biofuels marking an increase since 2015, while in Canada the share of oil remains almost stable.

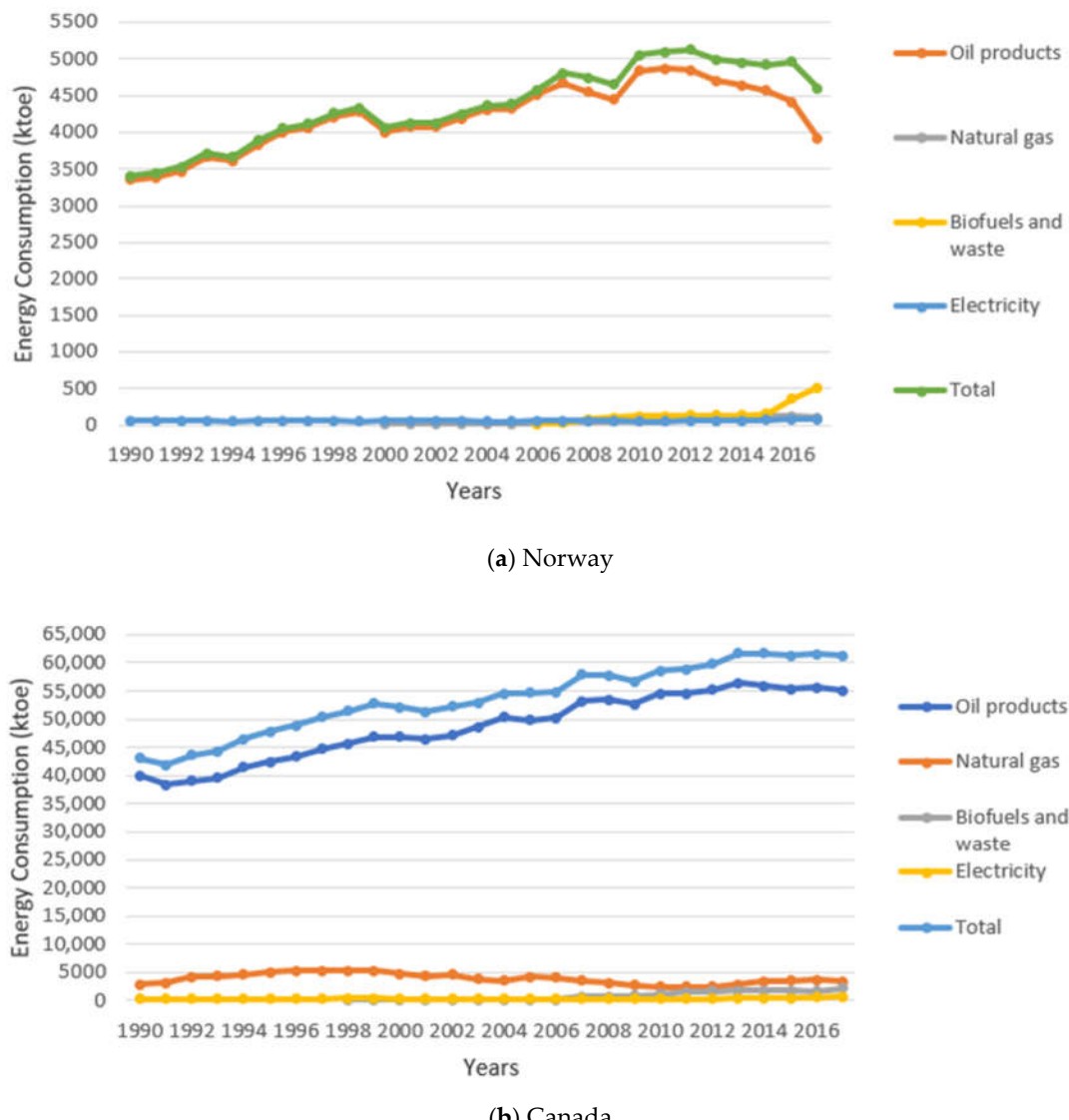

(**a**) Norway

(**b**) Canada

**Figure 3.** Transport energy consumption per energy source in (**a**) Norway, and (**b**) Canada. Source:
[16], own elaboration.

Regarding GHG emissions, the transport sector is one of the main contributors in both countries, accounting for almost 24%, while another major emitter is oil and gas extraction (dominating the Manufacture of Solid Fuels and Other Energy Industries category in both countries), surpassing the emissions from transport in Norway (Figure 4).

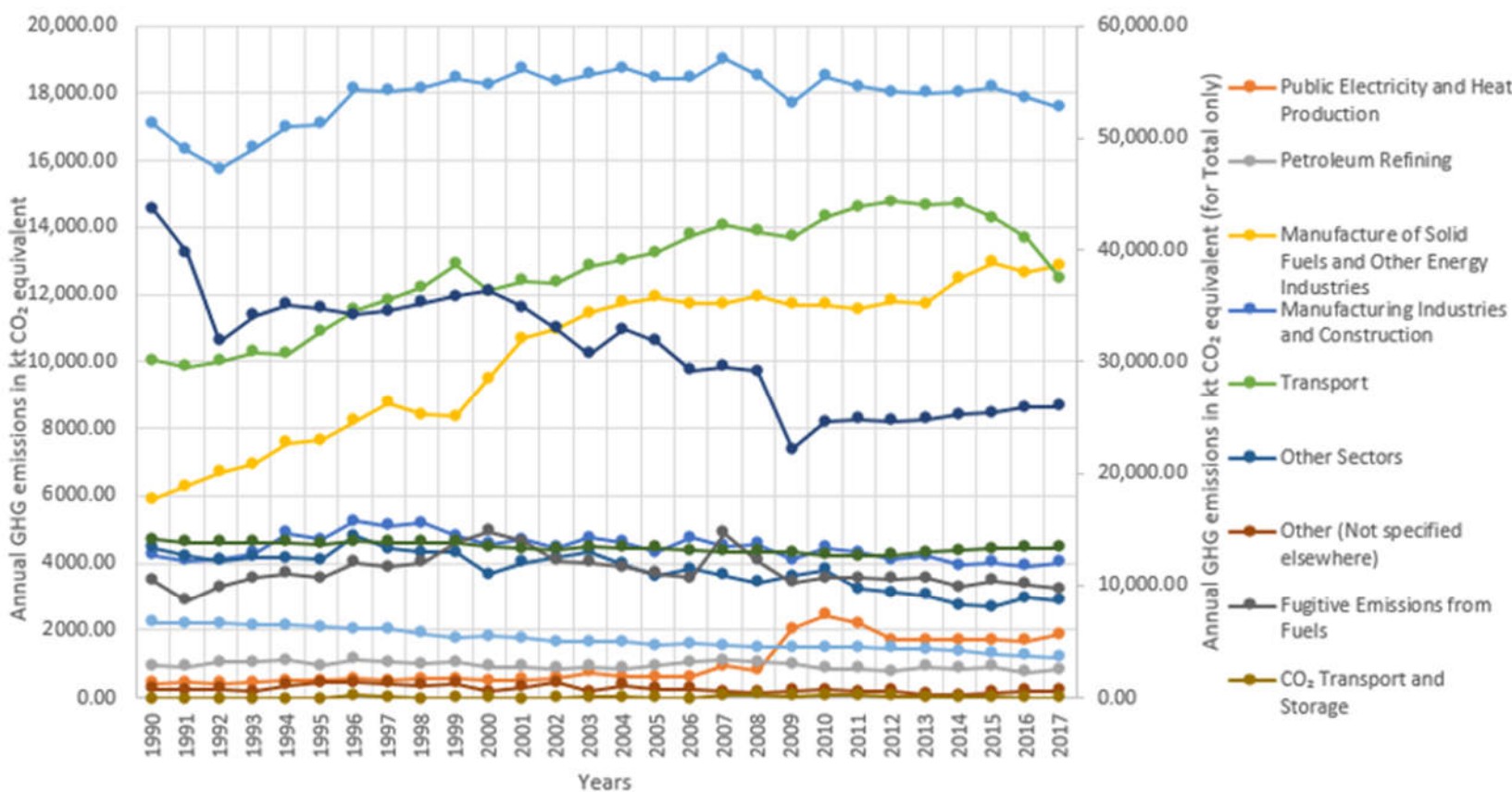

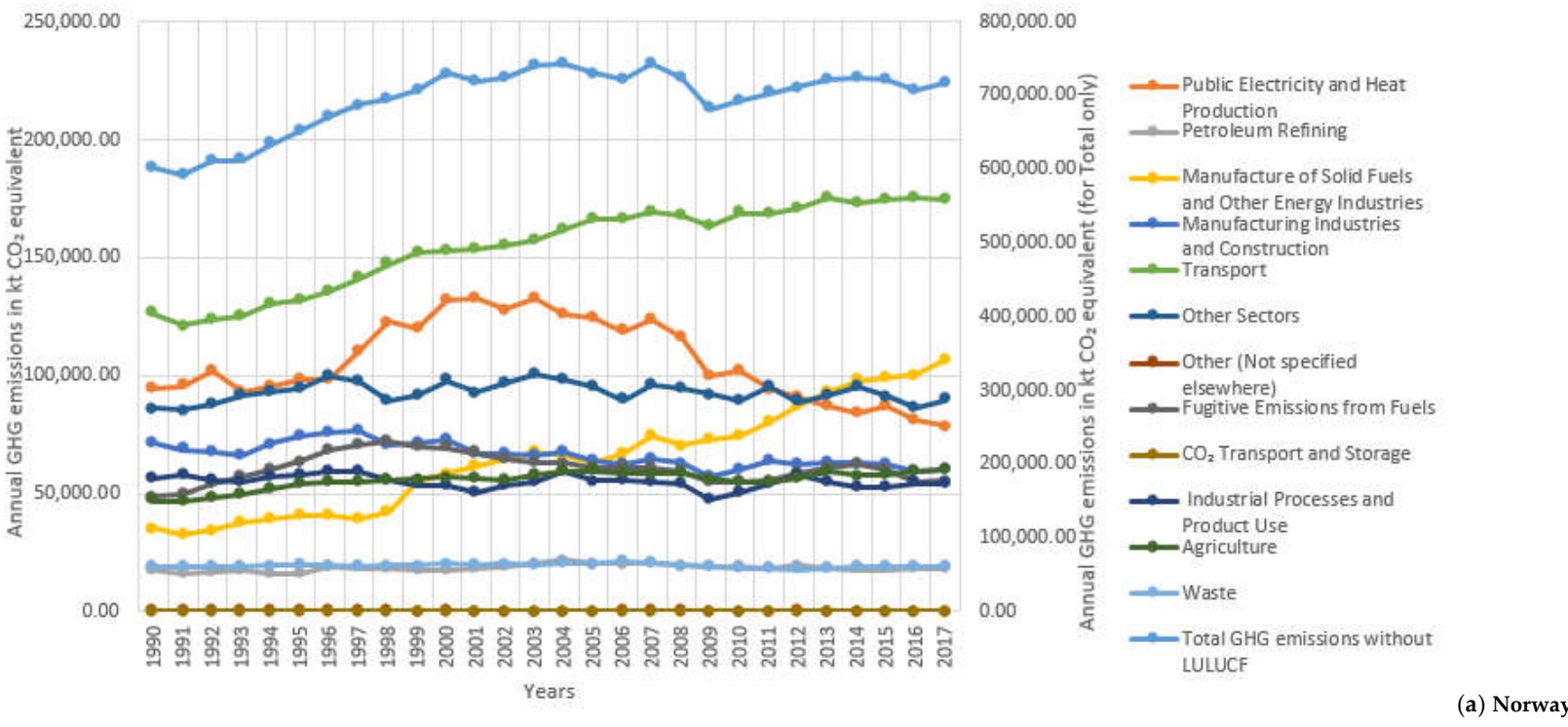

(a) Norway

(b) Canada

**Figure 4.** GHG emissions per sector in (a) Norway, and (b) Canada. Source: [16], own elaboration.

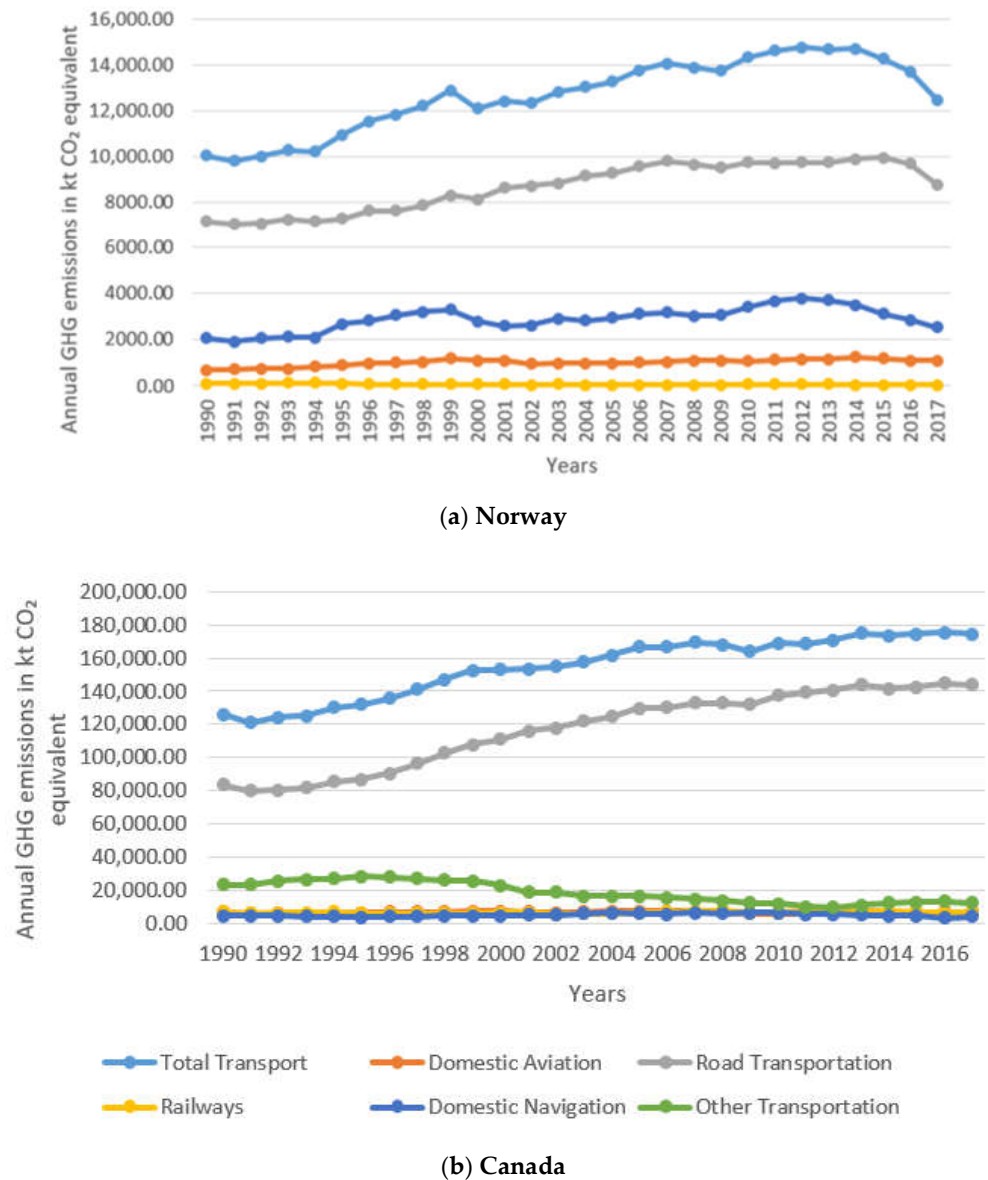

(**a**) **Norway**

(**b**) **Canada**

**Figure 5.** GHG emissions in transport per sector in (**a**) Norway, and (**b**) Canada. Source: [10], own elaboration.

In its 2015 Nationally Determined Contributions (NDC), Norway set a target for 40% of $CO_2$ equivalent reduction until 2030, compared to 1990 levels, in cooperation with the EU. In the following 2019 climate action pledge, a target of 35–40% cut in the transport sector, compared to 2005 levels [22], was established; in 2020, the country further strengthened its overall targets to 50–55% [23]. Regarding transport, emissions have been slowly decreasing since 2013 (Figure 4 and Figure 5)—the dip in 2017 was mainly caused by an increase in biofuels. Still, in 2017 the sector was among the top GHG emitters, with oil and gas extraction slightly trailing, and according to SSB data, transport emissions increased again from 2017 to 2018 [24]. Preliminary figures suggest that transport emissions dropped again in 2019: 7.7% in road transport (mainly driven by biofuels) and 6.5% in domestic sea and air transport [25]. Nevertheless, there is a need to take stricter measures to mitigate $CO_2$ emissions, while considering the country's ambition for all new passenger vehicles and light trucks to achieve zero emissions by 2025 [26]. Canada has also set targets, under the Paris Agreement, to achieve a 30% $CO_2$ emission reduction, compared to 2005 levels. Contrary to Norway,

Canada expects zero-emissions vehicle sales to have a 10% share, steadily reaching 100% by 2040 [27]. Neither country has made significant progress in decarbonisation, with Canada's total and transport GHG emissions reaching higher levels than in 2005, while Norway managed to slightly reduce its total and transport emissions by 5% and 6%, respectively, compared to 2005. This observation plays an important role when comparing policies between the two countries, since even in the case of Norway, progress is very limited compared to the country's pledges.

As far as carbon intensity is concerned, Norway has been showing a decrease in road transport since 2006, and in 2017 the carbon intensity of road consumption was 61.3 g $CO_2$/MJ; to put this into perspective, industry, which is another energy-intensive sector, had a carbon intensity of 24.9 g $CO_2$/MJ in the same year. Similarly, Canada's carbon intensity in road consumption has also been dropping since 2006, reaching 67.3 g $CO_2$/MJ in 2017, whereas in industry it equals 35.1 g $CO_2$/MJ. However, both countries' road transport carbon intensity remains lower than the global level (68 g$CO_2$/MJ) [16].

## 3. Methodology

Systems of Innovation frameworks are widely used to analyse the importance of a system's actors, networks and processes towards the technological change occurred in a specific system [28]. In particular, the SIS framework examines the interactions between actors and networks, as well as the impact of several factors on a sectoral level [9,11,29,30]. This framework has been a useful tool for a plethora of research papers examining various sectors. Indicative examples include the studies of Rogge and Hoffmann [31], Wesseling and Van der Vooren [32] and Rho et al. [33], which investigate case studies on power generation, the Dutch cement industry and the Chinese semiconductor industry, respectively. In comparison, our research focuses on the decarbonisation of the transportation sector of Norway and Canada, describes all sub-sectors of the transport sector (road, rail, sea and air) and is divided into four specific sections: "actors and networks", "institutions", "demand" and "learning processes and technologies", as proposed by Malerba [11]. The examination of these four factors is deemed important, since the technological progress of a sector is heavily affected by their co-evolution, which can play an important role in technological adaptation [34,35]. For instance, knowledge and technological processes provide important information on the ability of new environment-friendly technologies to penetrate the studied system, by examining the existent habits of the system actors. The SIS framework has been a dependable tool for much research in the transportation sector. For example, Thiel at al. [36] examined the decarbonisation of transport focusing on policy investigation rather than technological innovation; Chan and Diam [37] focused on policies regarding the Chinese transport sector; while Schade [38] studied the EU transport sector from various perspectives, including decarbonisation. Nonetheless, our approach differs from other studies by broadly capturing various systemic aspects of the transportation sector, with a focus on innovation progress and capacity as well as potential sectoral failures.

In this respect, we also integrate the SIS analysis with the SF approach [12], in order to assess and compare the potential challenges that each transport system encounters on its way to decarbonisation through the diffusion of innovative technologies. The SF framework examines potential barriers that may lead to a system's failure and separates them in four types: institutional, infrastructural, capabilities and interactions. This approach is considered essential for a better understanding of transition processes. The SIS framework alone may be able to include a broad picture of the innovation system and its structure, but it does not always suffice for the identification of specific barriers related to the diffusion of new technologies. Complementing it with the SF method can greatly contribute to the understanding of the examined systems.

In comparison with other studies, in which the SF framework is mainly used for the examination of industrial systems [39,40], our research conducts a comparative analysis of the transportation sector of the two countries. The main research questions posed during this comparison regard the structure of each county's sector, their differences as well as their weaknesses regarding decarbonisation and the diffusion of innovative technologies. The outputs of this analysis

may act as valuable policy insights for countries and policymakers to plan their future actions. Figure 6 presents the methodological approach followed in the study.

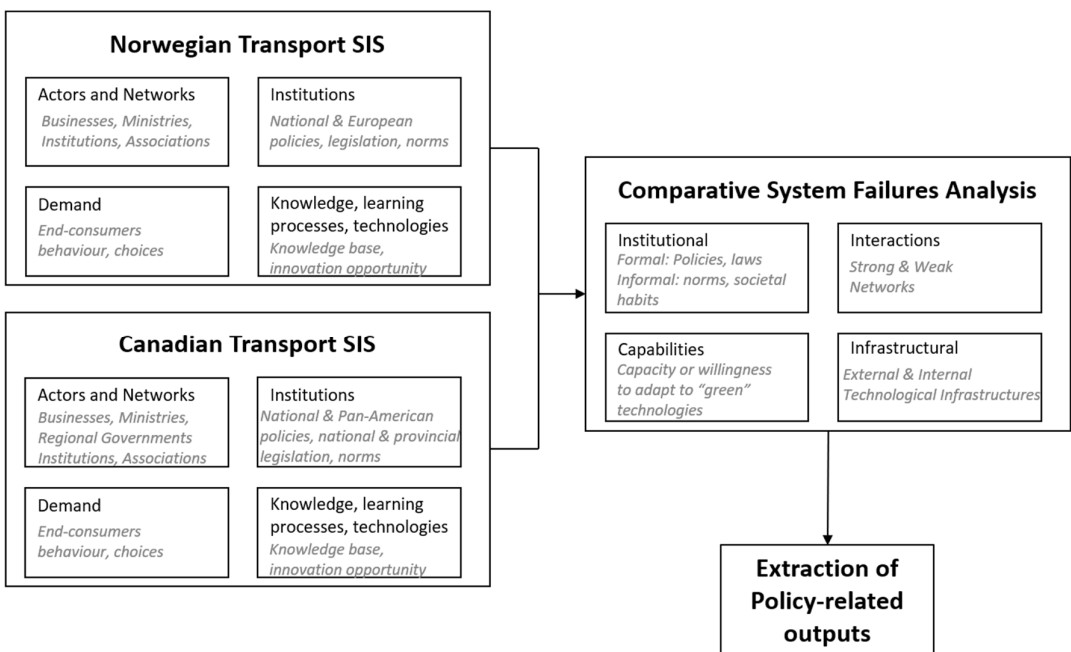

**Figure 6.** Methodological approach.

## 4. The Norwegian Transport SIS

### 4.1. Actors and Networks

The Norwegian transport sector constitutes a network of actors that cover a broad range of sub-sectors, including roads, rail, water transport (e.g., ferries) and civil aviation, operating under the governance of the Norwegian Ministry of Transport and Communications [41]. However, many of the policy instruments regulating transportation, such as taxes and regulations, are administered by other sector-overarching ministries, like the Ministry of Finance and the Ministry of Climate and Environment. Many administrative agencies operate under the Transport and Communications ministry. The Norwegian Public Roads Administration (NPRA) is responsible for the road transport system of the country, trying to increase security while implementing smart solutions. The management of the rail sector, after the Railway Reform in 2015, is shared between the publicly owned company Bane NOR, responsible for the railway network and traffic management, and the Norwegian Railway Directorate, that manages and coordinates the sector from a broader perspective, exploring potential developments. Similarly, the Norwegian Civil Aviation Authority (CAA) supervises all activities related to commercial air transport [42], while the Norwegian National Coastal Administration manages the administration of ports and seaways. The Norwegian Ministry of Trade, Industry and Fisheries is also active in the water transport sector, since the Norwegian Maritime Authority (NMA), which is subordinate to the ministry, is responsible for the ships flying the Norwegian flag and those on Norwegian waters, ensuring the legal protection and implementation of climate action policies.

As was the case with Bane NOR, the Ministry of Transport and Communications does not only rely on public agencies, but also operates a large number of government-owned companies. The Vy Group, formerly the Norwegian State Railways (NSB), constitutes one of the largest companies in the Nordic Countries, operating mostly passenger trains and bus services in Norway and Sweden, while the subsidiary company CargoNet offers environmentally friendly rail freight solutions. The operation of train lines has recently been subject to tendering, allowing competition. For instance, Go-Ahead Norway, a Norwegian subsidiary of the British company Go-Ahead, recently gained

some important stretches from Vy. Targeting the mobility and tourist sector as well, the company strives towards innovation, with a fleet of 250 electric cars. The financial strength of the company—along with other public companies in bus transport, like Boreal Norge AS and Tide ASA—limits private companies to more regional markets. However, to avoid these limitations, the private company Autolink AS, focuses on distributing automobiles through its subsidiaries Cargolink AS and Motorships AS, for railway and shipping, respectively. Avinor AS is another company owned by the Ministry of Transport and Communications, owning and operating the majority of state-owned airports. The company is a key actor considering sustainable transition, currently evaluating the electrification of civil aviation and identifying potential conflicts in collaboration with the Zero Emission Resource Organisation (ZERO), an environmental organisation working towards an emission-free transport sector, among other topics. Another association aiming for sustainability in transport is the Norwegian Electric Vehicle Association (Norsk Elbilforening), focusing on the promotion of electric vehicles as an environmentally friendly alternative to personal transport. As the largest consumer interest organisation in the Nordics, the Norwegian Automobile Federation acts as a stakeholder with significant influence, engaging in a broad range of transportation issues, including EV policies. The Ministry also owns a number of constructing companies, like Nye Veier AS (New Roads), which constructs, operates and maintains some major highways, as well as BaneService, operating mostly as a sub-contractor of Bane NOR. The combination of strong government-owned companies and private enterprises attests to the major role of the government in Norway's economy [43].

The petroleum industry is Norway's largest industry [44]. Large companies like Equinor ASA, one of the strongest operators in Norway regarding oil and gas [45], and Norsk Hydro, which focuses mostly on aluminium, may be considered as being outside the boundaries of the Norwegian transport SIS. However, acting at the borders of the system as major energy industry agents, they have the strength to formulate transition pathways affecting the decarbonisation of transportation [46].

On the research front, a major actor in the transport sector is the Institute of Transport Economics (TØI), an institution focusing on performing applied and academic research for all sub-sectors of transport, advising the authorities and disseminating key outputs. The institution collaborates with the University of Oslo on research projects and scientific cultivation. TØI is also part of the Nordic Road and Transport Research, which is a collaboration among the research institutions of the Nordic countries, aiming to effectively disseminate results among decision makers and assisting the knowledge transfer among the countries. Another major research institution in Norway is the CICERO Centre for International Climate Research, performing interdisciplinary research on climate change from a multi-sectoral approach, including transport [47].

*4.2. Institutions*

Norway is frequently considered as a pioneer in climate targets, and was one of the first nations to pledge to become carbon-neutral by 2050, back in 2008 [48,49]. However, despite the promising targets, carbon neutrality is not officially legislated, creating confusion regarding the substance of the original targets and the direction of Norwegian policy [50]. In fact, the Norwegian Climate Change Act, passed in 2017, sets general targets for Norway to become a low-carbon society by 2050, but does not specify that all cuts in GHGs shall occur in Norway [51]. Norway largely follows the European Union and participates in the EU Emissions Trading Scheme (ETS) and Effort Sharing Regulation (ESR) despite not being a member state [50,52].

In 2016, the National Transport Plan 2018–2029 [26] was revised in order to transform the transport sector to reduce emissions and costs, as well as to increase the safety and use of new technologies. A coherent strategy was necessary for Norway; as acknowledged in the plan, the transport sector, accounting for a quarter of the total GHGs, is mainly driven by the low population density. To finance the modernisation of transport, around €100 billion are to be used in the twelve-year period aiming to achieve a significant emission reduction of buses, coaches and trucks, with decrease rates varying from 50 to 100% until 2030, while also increasing the percentage rate of

biofuels in aviation to 30% by 2030 [53]. According to the plan, all new passenger vehicles registered in 2025 are expected to be zero-emission.

Norway is considered a leading electric vehicle (EV) country, with the world's highest share of EVs per capita. As early as 2014, more than 18% of new cars sold were electric [54], peaking at 60% during March 2019 [55]. Since then, this percentage has slightly decreased, with sales of new cars in 2019 ending at 42% EVs and 14% plug-in hybrids [56]. Nevertheless, this is the result of a long-term effort, since the state incentivised EVs from as early as 1990: EVs were exempted from import and value added taxes, an exemption that was made permanent in 1996; in 1997, they were exempted from road tolls, while in 2000 EVs were exempted from VAT as well [57]. These incentives managed to distribute the electrification costs between end-users and the state, with economic benefits for the entire economy [58]. However, the incentive policy faced some criticism, regarding the intervention in a single technology and certain privileges, like EVs being allowed to use bus lanes, causing traffic issues [59]. Such criticisms, combined with the increased diffusion of EVs—including premium brands and models—led to the phase-out of the incentives in the form of the 50% rule, which started in 2017, aiming to limit the incentives on ferry fares, public parking and toll roads to 50% of the price of fossil fuel vehicles [60].

To explain the strong EV policies in Norway, one must take into account the broader climate mitigation backdrop of the country. There is little gain in decarbonising the power sector, since Norwegian electricity is already 95% renewable. Climate action in Norway has always been internationally oriented, meaning that Norway is occupied with fulfilling its obligations to the UNFCCC scheme and the EU climate change mitigation framework. As seen in Figure 4, the petroleum and industry sectors constitute the top GHG emitting sectors in the country. These sectors are covered by the EU ETS, in addition to a general $CO_2$ tax (introduced in 1991), and, as such, already regulated in terms of GHG emissions. Since 1990, industrial emissions have been heavily reduced as a result of huge publicly financed R&D programmes. Conversely, emissions from petroleum extraction have increased in the same period (1990–2018) by over 70%, but cutting emissions from this sector is politically costly because of the central position of the petroleum sector in the national economy. Hence, political attention is turned to another emission-intensive sector—transport—where only aviation is covered by the ETS. According to the EU's Effort Sharing Regulation (ESR), Norway has to cut non-ETS emissions by 40% by 2030 compared to 2005 levels, and flexible mechanisms in the ESR framework are only to be used, according to the Cabinet, if "strictly necessary". In other words, ESR emissions should be cut domestically. Since Norway has very small emissions from buildings and waste, the agricultural sector already has an intentional agreement with the authorities to reduce emissions, and railways are largely run on renewable electricity; what is left is essentially road and sea transport. Road transport, in particular, is already heavily taxed; therefore, it is politically less costly to indirectly subsidise EVs through lower taxes than imposing more taxes and fees on road transport. For the same reasons, it is politically less costly to demand fuel retailers to blend in a certain share of biofuels into fossil fuels at the pump. The cost for taxpayers for the ambitious EV policies is, however, less visible in the public debate. In terms of sea transport, central actors see a considerable potential for economic growth in green shipping, and significant public R&D funds are therefore directed towards the maritime sector. However, the size of those funds is small compared to the R&D funds going into the petroleum sector.

### 4.3. Demand

Norway has an estimated population of around 5.3 million people as of 2019 [61] and is characterised by low density due to the large area covered, which subsequently leads to increased long-distance transportation [62]. However, the area of Greater Oslo has more than a million inhabitants, creating a complex system with both urban and rural elements, concentrating half of the public transport activities [63]. This sparse population pattern, coupled with the long coastline and the fact that many people and businesses are located near the sea, requires a variety of transport methods to be used on the demand side, which is also evident from the distribution of emissions

seen in Figure 5, where despite the dominance of road transport, other means like water and air transport contribute non-negligible shares.

In 2018, there were 2.7 million registered private vehicles, of which 7% were electric (0.2 million cars) [64]. In 2019, there were 2.4 million households in Norway [65], meaning that, on average, each household owns one vehicle. In 2018, passenger cars covered almost 58 billion km, which is around 78% of the 74 billion km travelled by all road vehicles, including buses and lorries [66].

Domestically, passenger transport in 2018 accounted for 84,445 million passenger-km, while the transport of goods amounted to 53,081 million tonnes-km, of which 26,706 include mainland transport, while the rest refer to transport in the Norwegian continental shelf [67]. Figure 7 presents the share of each sector both in passenger and goods transport. Even though an increased percentage of maritime transport is observed for freight, the majority of goods require short-distance distribution [68], further establishing the dominance of road transport.

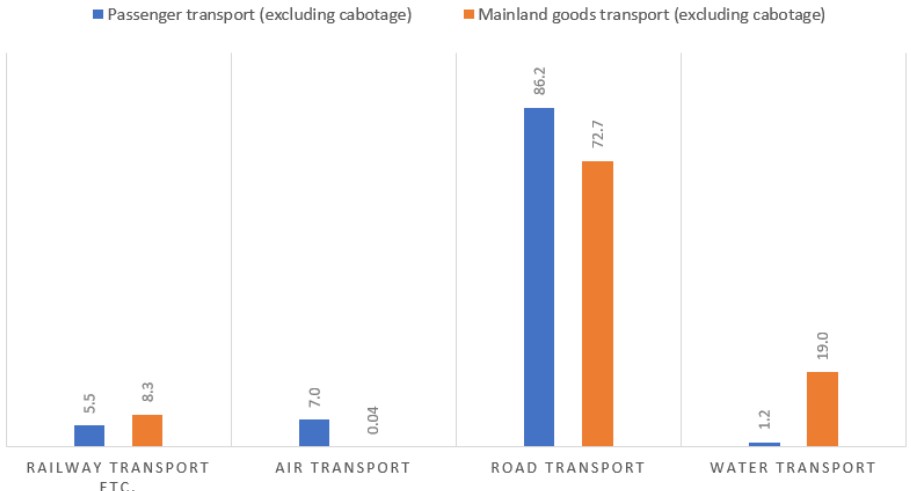

**Figure 7.** Shares of passenger and goods transport domestically in 2018. Source: [67], own elaboration.

The ability of railway transport to convert passenger transport to revenues, accounting of around €500 million provides the sub-sector with the capacity to effectively finance innovation towards sustainability, as illustrated in Figure 8.

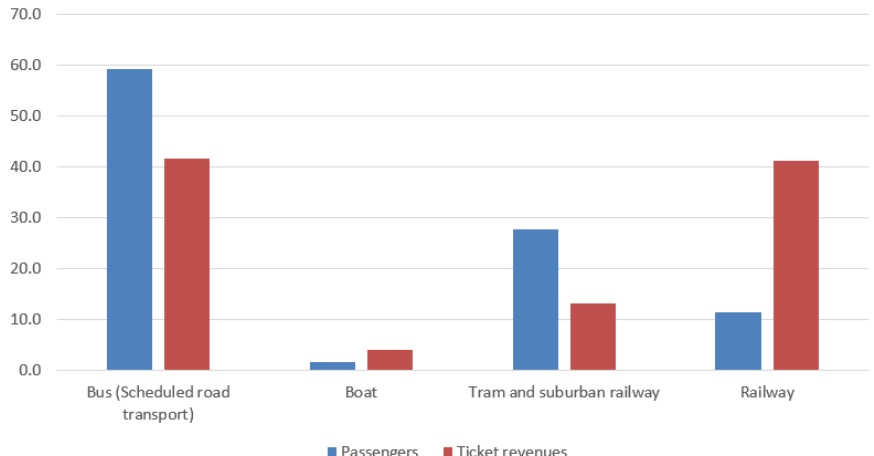

**Figure 8.** Shares of public transport passengers and ticket revenues in 2018. Source: [69], own elaboration.

### 4.4. Knowledge, Learning Processes and Technologies

The existing technological base of Norwegian transport relies heavily on traditional fossil fuels across all individual sub-sectors. However, the increased production from renewable energy renders transport electrification an efficient pathway for the sustainability of the SIS. Initiatives throughout the broad spectrum of transport are in place, even though they are still premature, indicating the actors' positive attitude towards the process [60].

Norway is one of the leading countries regarding the diffusion of EVs [54], having already initiated the transition towards sustainability in mobility. The country's electricity mix relies mostly on hydropower, with a share of at least 95% (Figure 2) that could even reach 99%, depending on the yearly conditions [70]. Clean electricity production cultivates ideal conditions for climate action by means of electrification, as regions in which electricity powering EVs is sourced from GHG-intensive sources, like coal or oil, reduce or negate electrification benefits [71]. Norwegian households also rely on home charging, since 75% have their own dedicated parking [72], counterbalancing the negative infrastructural effect caused by the small ratio of fast chargers compared to BEVs in stock [73]. To build this infrastructure, a public support scheme has taken place as early as 2009, while the fast-charging infrastructure of Norway was supported mainly by local utility companies, that initiated the first round of support for the operation of fast-charging stations [74]. Despite the favourable conditions, EV diffusion is still at an early stage, as shown in Figure 9, presenting the fuel mix used in private cars in 2017.

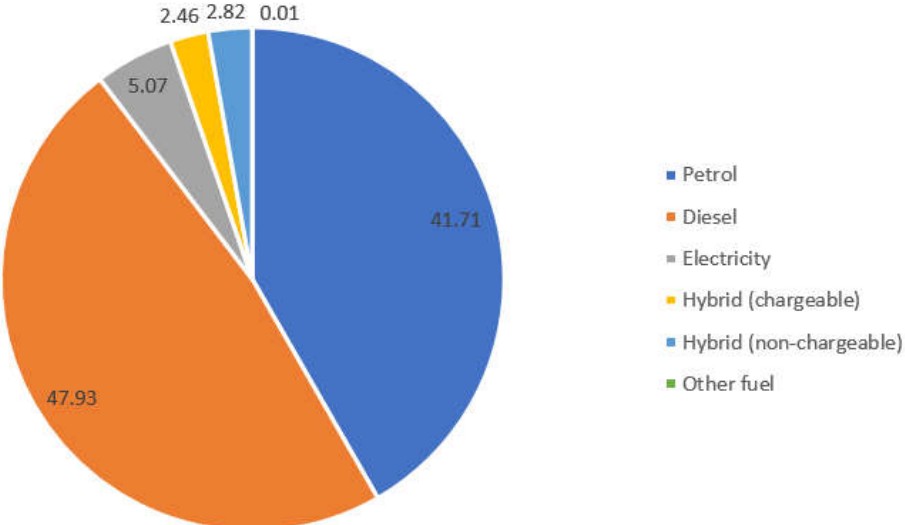

**Figure 9.** Shares of fuel used on private cars in 2017. Source: [64], own elaboration.

Fossil fuels dominated in private cars in 2017, with diesel-powered vehicles constituting almost half of the fleet. The energy content per litre is higher for diesel than for petrol [75], limiting the margin for efficiency provided by EVs and showing that increasing the fuel efficiency of vehicles will require wider changes in the transport industry [76]. However, the share of electric cars among the total registered cars is progressively increasing, reaching almost 7% in 2018 and 9% in 2019 (Figure 10). The distance covered by EVs, though, during the same period, was 5.3% of the total, surpassing 7% a year later, in 2019, with EVs covering 3400 million km out of a total of 45,562 million km [77]. The use of biofuels has also increased in recent years (Figure 3), but the production cost has led to high levels of imports [78]. The controversy surrounding biofuels [79], especially regarding the unsustainability of palm oil as a source [80], coupled with policy inconsistencies related to incentives, has limited the diffusion of biofuel innovation in the Norwegian system [81]. Despite the ambitious targets set for electrification, biofuels constitute an important additional measure as part of Norway's decarbonisation strategy for the transport sector.

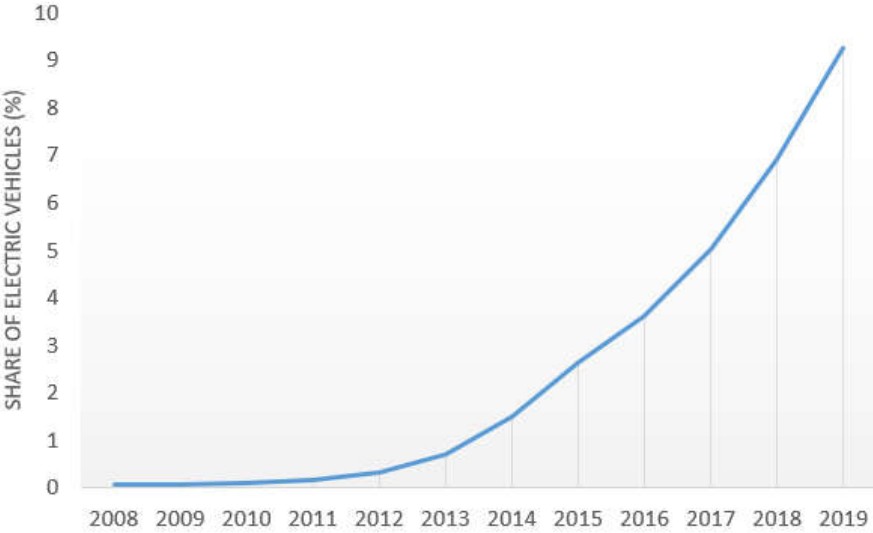

**Figure 10.** Share of EVs in Norway. Source: [82], own elaboration.

Regarding water transport, efforts are also focused on limiting the use of fossil fuels in ferries and passenger boats, but also in other means of transport, like cargo trucks, by introducing alternative fuels [83]. Despite EV diffusion being the main innovation in the Norwegian transport system, the maritime sector also has a significant share in the transport of goods, as seen in Figure 7. This is also evident from the share of emissions in which road transport is dominant, but significant amounts are also produced by sea transport, as seen in Figure 5. While Norway has no car industry, its maritime sector is technologically cutting-edge from an international perspective, as exemplified by the production of the first fully electric ferry in 2015 by the Norwegian company Norled and the country's leadership in the use of electric ships in the following period [84]. In Norway's 2015 NDC, along with the ambition of reducing emissions in the transport sector, environmentally friendly shipping was mentioned as part of the priority areas of the country's national policies, indicating a significant political focus on the potential for green growth in the maritime industry. In the government's action plan for green shipping [85], multiple funding agencies were described that could provide financial support, like Enova, whose role as an important funding agency would be maintained in the following period, "Innovation Norway" with green shipping being a major recipient of almost €7 million, and the MAROFF programme from the Research Council of Norway, with the Ministry of Trade providing almost €16 million for maritime research and innovation in 2017. The collaboration between the authorities and the business sector is also important, with public-private partnerships like the Green Shipping Programme receiving more than half a million euros from the national budget in 2019. In 2016, the National Transport Plan 2018–2029 pledged that by 2030, 40% of all ships in short-distance sea shipping will use biofuels or be low- and zero-emission vessels, that all new national highway ferries will use low- or zero-emission solutions and that county municipal ferries and fast ferries will use low- and zero-emission solutions [26]. Green shipping is also part of the Norwegian stimulus packages to boost the economy in the wake of the Covid-19 breakout.

On a similar approach in the aviation sector, Avinor acquired a small electric airplane to trigger research on electrifying air transport [86], although progress is limited compared to marine transport.

In the Norwegian railway network, 62% of lines were electrified as of 2007 [87], while as of 2019 more than 1000 km of non-electrified lines existed [88], mostly covered by fossil fuels. Electrifying the remaining parts of the network or introducing more efficient fuels is a major challenge of the transport SIS in Norway, because of the costs and a sparse population pattern.

The key outputs from the Norwegian transport SIS are presented in Table 1.

**Table 1.** Key outputs of Norwegian transport using the SIS blocks.

| SIS Blocks | Key Results |
|---|---|
| Actors and Networks | • High participation of the public sector either through state-owned companies or private-public collaborations<br>• Strong actors from the petroleum industry |
| Institutions | • Early policy through incentives for EVs, which has been criticised in recent years. |
| Demand | • High share of road transport<br>• Water transport shows a significant percentage in the transport of goods |
| Knowledge, learning processes and technologies | • Current regime dominated by diesel and petrol vehicles<br>• EVs remain the major innovation in the system, with their share increasing<br>• Electrification is the main measure of the decarbonisation strategy, but the Norwegian government is interested in biofuels as an additional measure<br>• Local utilities responsible for developing the fast-charging infrastructure |

## 5. The Canadian Transport SIS

### 5.1. Actors and Networks

The Canadian transportation sector is characterised by a variety of activities depending on the means used. Each transportation method is related with different associations, networks and firms. Regarding automobiles, Canada is considered one of the biggest car producers worldwide, manufacturing more than 2 million vehicles per year [89]. Although Canada is a major car producer, there is no automobile manufacturer headquartered in the country: all stakeholders are firms established in other countries, especially in the USA and Japan (e.g., GM and Toyota) [90]. Canada also hosts one of the most important bus manufacturers worldwide and the largest in North America, the NFI Group, as well as one of the major train and plane manufacturers, Bombardier Inc. Furthermore, because of its abundant fossil fuel reserves, Canada is also strongly involved in oil and natural gas production, especially in the Western provinces [91]. There are Canadian as well as international fossil fuel companies operating in Canada. Regarding the former, Suncor Energy and Canadian Natural Resources are among the biggest in the country [92]. International fossil fuel corporations are also quite active in Canada, like Chevron and Exxon Mobil. The activity of fossil fuel companies is an important factor in the system, since they are usually involved in political campaigns to promote their actions [93]. This can act as a barrier against rapid decarbonisation actions, considering that the energy industry is a large contributor to the Canadian economy [94].

Apart from vehicle manufacturers and fossil fuel producers, passenger transport operating and logistics companies also play an important role in the setup of the Canadian transportation sector and its low-carbon technological transition. Each town and city in Canada has its own urban bus transportation network, while the three biggest cities (Toronto, Montreal and Vancouver) also possess a metro system. Furthermore, some smaller cities (like Calgary and Edmonton) have a light rail system for urban transportation [95]. This plurality of independent networks and associations can lead to a greater need for cooperation and coordination, which is not easily achievable, resulting in delayed actions towards the transition to a "greener" system.

Apart from urban transportation, Canada is characterised by a variety of bus and train companies operating long-distance routes, being the second largest country in the world. Regarding freight transport, Canada hosts 70 public and private rail companies and approximately 208,000 trucking businesses [96]. Regarding sea and air transportation, there are fewer firms operating in Canada, and the majority of them are headquartered in other countries, especially for international transportation [96]. The only exceptions are the Air Canada (passenger and freight), WestJet

(passenger) and Air Transat (passenger) airlines, which accommodate many international destinations [96].

As discussed above, transportation is a complex sector, as it consists of many sub-sectors, such as road, rail, sea and air transport, all of which can also be separated into passenger and freight. It is therefore supported by many associations affecting the sector's landscape and the diffusion of new technologies. At the local level, the CUTA (Canadian Urban Transit Association) plays an important role in knowledge diffusion, urban authorities' advocacy and the communication among various local transit authorities. It consists of hundreds of members, such as universities, federal and provincial ministries, businesses and urban transit authorities. Some significant members of this association are the Transportation Research Institute of the University of Toronto, the Transportation Association of Canada (TAC) and corporations like the NFI Group. TAC, in particular, is a similar umbrella organisation, but with a broader spectrum of actions, focusing on road transportation and infrastructure at the national and local levels, while its pillars of action include environmental protection and the mitigation of climate change: it has established its own Environment and Climate Change Council, which proposes technical solutions for both mitigation and adaptation (TAC). Similar umbrella organisations for other means of transport are the Railway Association of Canada (RAC), the Shipping Federation of Canada and the Air Transportation Association of Canada (ATAC). There are also associations representing freight transportation stakeholders. The Private Motor Truck Council of Canada is the main Canadian association promoting the interests of the private truck companies and disseminating knowledge on various issues related to road transport. There are also associations representing companies purchasing freight transport services and contributing to the interaction between freight transportation companies and their customer base; the Freight Management Association of Canada (FMA) is considered an important federation among these, with its members' annual contribution to the Canadian economy amounting to $100 billion dollars, and the transportation services they use to $4 billion dollars.

Apart from numerous associations, there are also federal and public actors involved in the transportation sector, especially in the diffusion of new "green" technologies. An important actor of knowledge "production" and diffusion is the Transportation Research Institute of the University of Toronto, which engages in relevant research areas such as alternative fuels, traffic congestion, etc. [97]. Another university carrying out research in transport—and particularly in EVs—is the University of Concordia in Quebec, authoring a high number of electric mobility-related publications, while the Hydro-Quebec Research Institute, owned by the public utility Hydro-Quebec, excels at research focused on electric technologies, such as batteries [98].

Federal government institutions are also related to the sustainable, low-carbon transition of the country's transportation sector. Transport Canada is the ministry related to regulations regarding this sector, by promoting policies towards its safe, efficient and environment-friendly operation. Environment and Climate Change Canada, the ministry related to climate change and action, also plays an important role, being responsible for aspects of global environmental change and air pollution that are intertwined with the transport sector [99]. Finally, in 2015, the Canadian Council of Ministers of the Environment (CCME), including the ministers of environment and climate change of each Canadian province as well as of the federal minister, established a Climate Change Committee (CCC), aiming to implement the Pan-Canadian Framework for Clean Growth and Climate Change (PCF), which sets various targets for the entire economy [100].

*5.2. Institutions*

Among the economic sectors targeted by the PCF, special attention is given to electricity, industry, agriculture and transportation. Regarding the latter, environmental policy is broken down into four pillars: setting stricter standards and increasing energy efficiency; raising the percentage of zero-emission vehicles; investing in public transit and infrastructure; and using cleaner fuels. An example of the implementation of this framework is the electrification of transportation in Quebec, which aims to have 100,000 electric vehicles until 2020 [100]. This is to be achieved with the

allocation of at least $6.2 billion towards climate change mitigation with an emphasis on electrification, as illustrated in the latest budget of the province [101]. The electrification of transportation in Quebec, a province featuring an electricity system that is heavily dependent on hydro, is considered a significant measure towards the abatement of GHG emissions [98].

Another important measure taken by the Canadian government is the legislation of the Greenhouse Gas Pollution Pricing Act (GHGPPA) in late 2018. The Act established charge rates for fuel use in various sectors, including the transportation sector, which are dependent on the emissions produced by the combustion of each fuel and are different in each province. The GHGPPA is in force in half of the provinces and two territories: Ontario, New Brunswick, Manitoba, Saskatchewan and Alberta; and Yukon and Nunavut [102]. The legislation concerns fuel distributors, fuel producers and specific road, rail, marine and air carriers, who have to register on a platform of the Canada Revenue Agency (CRA) in order to claim their use of fuel, which is charged according to its associated emissions [103]. This results on increased costs for transport stakeholders, including people that are not professionally involved in this sector, since the price of fuels on gas stations has already been increased [104].

Another important factor of environmental law is Canada's sub-national division: the country is divided into three territories and ten provinces with their own legislation, which may be less or more proactive than federal legislation [17]. For example, the Quebec and Alberta provinces have legislated their own carbon pricing regulations in 2007 and 2015, respectively, acting faster than the federal government [105,106]. On the other hand, the province of Ontario, although having established its own carbon pricing scheme since 2017 [105], showed great resistance to the proposed GHGPPA of the federal government after the provincial election of the conservative party. In fact, the provincial government of Ontario initiated legal processes disputing the constitutionality of said Act, but the Court of Appeal of Ontario adjudged that the GHGPPA does not violate the Canadian constitution. There was a similar case in the Court of Appeal of the province of Saskatchewan, resulting in a similar verdict [107]. Despite their result, these appeals indicate that provincial legislation and policy may not always benefit climate action, leading oftentimes to conflicts with federal policy; while in other cases, sub-national environmental and climate policy may act faster than the federal government, with implications for the transport sector.

These carbon pricing schemes constitute an effort of some Canadian provinces to participate in an emission trading scheme organised by the West Climate Initiative (WCI) and comprising the state of California (USA) and the provinces of Quebec, Nova Scotia and British Columbia. Membership of this scheme varies with time: Ontario, for example, became a member of this initiative since almost the beginning but withdrew its membership in late 2018, after the election of its new provincial government, which proposed the Cap and Trade Cancellation Act [108].

Transportation 2030 is a policy of Transport Canada aiming to transform Canada's transport sector into a more efficient, safer and greener sector by 2030. The policy's budget for 2017 was higher than half a billion dollars and consisted of numerous measures: it included $120 million for electric vehicles and alternative fuel investments, $56.9 million for GHG regulation development and $229 million to support the Clean Energy and Clean Transportation Innovation Programming [109]. One of such initiatives is the establishment of a research fund supporting rail, marine and air transportation, so that they transition towards a more environment-friendly operation. This research fund cooperates with various public and private organisations operating in academia, government, transport and other relevant sectors [110].

*5.3. Demand*

Canada is the second largest country worldwide and two of its three largest metropolitan areas, Toronto and Montreal, are located on the east side, while Vancouver is situated on the west coast of the country [111], at a distance of approximately 4400 km by road and 3400 km by airplane from Toronto. Therefore, Canada is characterised by extensive transportation needs. Specifically, transportation accounts for 5% of its GDP, and Canada has one of the largest transportation sectoral GDPs per capita [112].

The country has a very wide road network, with a length of more than 1,300,000 km, of which 443,000 are paved [113]. It is noteworthy that the transportation activity related to road transport is increasing in every sub-sector, leading to the emissions from road transport not only dominating in this sector, but also steadily increasing, as seen in Figure 5. In the 2000–2013 period, the passenger-km travelled by cars increased by 13% [114], while gasoline sales increased by 10% in the last ten years [115], although cars' engine efficiency was constantly enhanced [116]. The increase of passenger transportation is also visible in the bus industry, the revenue of which increased from 8.6 billion dollars in 2005 to 20.3 in 2017 [117]. A similar effect was also observed in freight transport, where the activity of trucks doubled between 2000, with 84.7 billion-tonne km of domestic activity, and 2014, with 166.6 billion-tonne km and an upward trend. Moreover, the tonnage of cargo transported in the same period almost tripled [118].

The vastness of the Canadian land has also resulted in an impressive rail network, the length of which is more than 60,000 km [113]. In comparison with the road sub-sector, the increase in rail transportation was slightly inferior: the total passenger car-km increased from 58 billion in 1990 to 71 billion in 2017, fluctuating significantly during this period [119]. Furthermore, the domestic freight activity increased from 207 billion-ton km in 2000 to 282.2 in 2015 [111]. On the other hand, the total freight activity (including international trade) of the train sector has shown a similar augmentation rate, from 534 billion-tonne km to 761 billion-tonne km. This increase was also reflected in the revenue of rail companies, which almost doubled in this period, reaching $13.52 billion dollars in 2014 [119]. Although rail activity increased, the total fuel consumption (diesel oil) remained almost stable during this time span [120].

Another important means of transportation, although less extended in Canada, is water transport, divided into three categories: inland waterways, referring to rivers; the Great Lakes, referring to the boats traversing these lakes; and sea transport. The length of inland waterways and the marine network of the Great Lakes is 2825 and 2662 km, respectively [113]. Furthermore, the country has 557 port facilities [96]. Regarding the inland waterways and the Great Lakes, domestic activity fluctuated during the 2000–2011 period but did not show a clear upward or downward tendency, with an annual average of 4.7 and 23.83 billion-ton km, respectively. On the other hand, domestic coastal shipping greatly increased during the same period [118]. It is important to mention that most sea trade is related to international trade; in 2011, the tonnage traded internationally accounted for 73% of the total sea freight activity [121]. Another important sector of marine transportation is the containership sector. The main ports in Canada are Vancouver, Montreal and Halifax. From 2005 to 2011, their total activity increased by almost 400,000 twenty-foot equivalent units (TEUs). In 2011, the number of TEUs handled in these three ports was 2,500,000, 1,220,000 and 370,000, respectively, with the port of Halifax showing a downward tendency. Another fact that demonstrates the increased number of containers handled in Canadian ports is the rapid growth of the Prince Rupert port in British Columbia, which started operating in late 2007 and by 2011 had surpassed the port of Halifax, handling 400,000 TEUs [122]. Last but not least, marine transportation also serves travelling passengers. In 2018, the major ferry routes in Canada carried 53 million passengers and 21 million cars [96].

The last method of transportation is aviation. Canada has approximately 1200 airports [113], serving passenger and/or freight flights. The four busiest airports of Canada are located in Toronto, Vancouver, Montreal and Calgary, with Montreal having two airports operating freight flights [123,124]. In the period of 2008–2018, passenger activity significantly increased, from 109 million to 159 million passengers. In fact, domestic passengers increased by 23 million and transborder passengers between Canada and the USA by 10 million, while the rest of the increase is related to other international activities [124]. This rapid increase is also reflected on the total activity measured in passenger-km, which increased by almost 20% in just two years (2015–2017) [125]. A similar upward tendency is also observed on air freight transport. In 2008, a total of 972 million tonnes of cargo were handled, whereas in 2018 this figure reached 1434 billion tonnes. Domestic cargo increased by 200 million tonnes and international cargo, excluding transborder trade, by 275 million, while transborder freight slightly decreased [123]. This is reflected on the total tonne-km handled,

which increased by 38% during the 2015–2017 period [125]. As expected, the increase of air transportation activity is reflected on the aviation's fuel consumption, which in 2017 had increased by 14% compared to 2012. Specifically, in 2012, the total fuel consumed was 6.6 billion litres, and in 2017 it reached 7.55 billion litres [126].

Demand demonstrates that Canada has a fast-growing transportation sector, whose GDP has increased from $58 billion dollars in 2000 to $90 billion dollars in 2019 [127]. This constant growth in transportation has a two-fold impact on its transition to a sustainable sector. A growing sector is capable of investing in new, low-carbon or carbon-neutral technologies, with a lower degree of dependency on state subsidies [128], since incumbent firms are growing and able to invest more. On the other hand, a growing sector demonstrates a high-energy demand, and thus stricter policies must be implemented in order to avoid a carbon lock-in [129].

*5.4. Knowledge, Learning Processes and Technologies*

As mentioned before, the transportation sector consists of a plethora of sub-sectors. Some of these are characterised by a variety of fuels and others are dependent on a single fuel. Road transportation is divided into passenger and freight. Regarding passenger transportation, various types of vehicles are used. The most common are vehicles fuelled by gasoline, accounting for almost 94% of new registered passenger cars between 2014 and 2018. Furthermore, almost 4% of the cars registered in the same period use diesel as their energy source. The remaining 2% consisted of hybrid, plug-in hybrid and battery electric vehicles. Nonetheless, it is noteworthy that the sales of electric cars have recently met a significant increase. In 2018, the newly registered electric cars reached 3.5% of the total cars, including hybrid and plug-in hybrid cars [130]. Apart from the fuel used on each car, another important factor is the fuel efficiency, depending on the equipment and size. Canada is considered the world's least fuel-efficient country regarding passenger vehicles, since it has the largest and second heaviest cars worldwide. The average emissions of a Canadian light-duty vehicle per kilometre are higher than 200 g of $CO_2$ [131], which attests to a high potential for improvement. There is a similar trend in neighbouring countries like the USA [132]. On the other hand, road freight transportation is characterised by the exclusive consumption of diesel oil by trucks handling cargo [133]. Lastly, regarding road transport, biofuels are mixed with diesel and gasoline in order to reduce their carbon emissions [134]. The mix rate of biodiesel and ethanol in diesel and gasoline, respectively, varies per province, in accordance with federal regulations. The maximum rate for ethanol is 8.5%, in the province of Manitoba, whereas the maximum rate for biodiesel is 4%, occurring in the provinces of Ontario and British Columbia. However, there are provinces such as Quebec that do not mandate a specific rate percentage [135].

Regarding train transport, the main driver of fuel consumption is freight, which uses diesel as its sole energy source. The same phenomenon is observed in intercity passenger trains. The only other energy source used in rail transportation is electricity, mainly used for urban transit. Indicatively, in 2013, 93 PJ of diesel were used for intercity passenger and cargo transportation, whereas only 3 PJ of electricity were consumed for urban transit during the same year [136]. Air transportation is characterised by a similar distribution of fuels, with two different fuel types being mainly used: jet fuel and aviation gasoline. Jet fuel is dominating this sub-sector, with a usage percentage above 99%. Lastly, marine transportation is also dependent on two types of fuels, but these two fuels are more homogenously distributed: residual fuel oil accounts for 65% of energy consumption, whereas distillate (diesel) fuel oil for the remaining 35% [136]. Contrary to road and rail transportation, aviation and marine transport do not use biofuels in their fuel mix, as is the case worldwide [137].

There are many technologies that can be used in order to reduce GHG emissions in the transportation sector, towards climate change mitigation. Some of them can be applied to specific sub-sectors, whereas others can be widely used. Widely applicable technological improvements include the increase of electric vehicles for road and rail transportation, as well as the introduction of electric mobility in air and marine transportation. Regarding passenger vehicles, some Canadian provinces (Ontario, Quebec and British Columbia) have provided subsidies to citizens in order to

promote the purchase of electric vehicles. Likewise, the federal government also legislated in 2019 the provision of subsidies (up to $5000) to people buying electric vehicles [138].

Another area in which electric vehicles are not widely used in Canada is train transportation, which is dominated by diesel combustion, since the electrification of train services leads to financial uncertainty, especially for long distances between different regions [139]. Furthermore, electric mobility is considered a future solution for the reduction of GHG emissions in road freight transportation, by means of electric trucks [140], as well as in marine and air transportation, with electric boats and planes [141,142]. These technologies have been examined mainly on a research level and have no major applications. On the other hand, the increased penetration of electric cars and trains, which is already taking place in other countries, would result in significant GHG emissions reductions because of Canada's hydro-dominated power generation mix [143]—especially in provinces like Quebec, with a high share of production from renewable sources. Therefore, the only sector that has experienced a mild transition towards a reduced dependence on fossil fuels in Canada is road transportation, because of the slow penetration of electric cars. The other sectors have not demonstrated any noteworthy tendency towards the exploitation of Canada's clean electricity.

Another fruitful action is the increase of biofuels in diesel and gasoline blends, by promoting advanced biofuels [134]. Specifically, Canada has the largest biomass reserves per capita, accounting for 7% of the global potential biomass production [144]. Another fuel that can be used for transportation needs is green hydrogen, through electrolysis using RES. Furthermore, the country is characterised by an important research activity in hydrogen exploitation, regarding its production, storage and usage on the transportation sector [139,145–147]. Canada has a lot of hydrogen stakeholders, ranging from hydrogen production companies to car manufacturers, that are interested in hydrogen fuel cell technology.

The key outputs from the Canadian transport SIS are presented in Table 2.

**Table 2.** Key outputs of Norwegian transport using the SIS blocks.

| SIS Blocks | Key Results |
|---|---|
| Actors and Networks | • Mostly private firms operate in the sector<br>• Fossil fuel powerhouses have significant influence<br>• Increased localised elements. Western provinces are strongly involved in oil and natural gas production |
| Institutions | • Policy depends both on the federal and the regional governments<br>• Some provinces resist federal legislation (e.g., GHGPPA)<br>• No universal carbon pricing scheme. Some provinces participate in a scheme with the state of California (US) |
| Demand | • The vast area of Canada creates increased transportation needs for multiple means |
| Knowledge, learning processes and technologies | • The current regime is dominated by inefficient gasoline vehicles<br>• EV share slowly increases<br>• EV diffusion more influential in regions with a high share of hydro-power generation (e.g., Quebec) |

## 6. Comparative Analysis

The transportation sector in both Norway and Canada is characterised by the dominance of fossil fuel combustion across all sub-sectors. This means that this sector is facing a contingent carbon lock-in related to fossil fuels and, especially, oil products such as gasoline and diesel. This phenomenon is caused by multiple factors, including the high capital costs [148], leading the majority of transportation stakeholders (from car manufacturers to car users) to stick to the use of conventional fuels. As Klitkou et al. [149] note, the existing fossil fuel economy of scale in the

transport sector implies a technological lock-in; the risk in both countries is high, acting as a significant barrier to a low-carbon transition. In this section, we compare the transportation system of two countries (Norway and Canada), in order to detect and analyse these barriers. The ability of their systems to overcome said barriers is studied through the SF framework [12], by examining specific features of each innovation system.

### 6.1. Institutions

Institutions can be separated in formal and informal. Formal institutions are related with the broader legal and regulatory system, with a focus on specific regulations. On the other hand, informal institutions concern social values, norm and habits [150]. The failures associated with these types of institutions are characterised as hard and soft institutional failures, respectively [151].

At first glance, Canada seems more susceptible to institutional failures. Delving further into the formal institutional framework, both countries have taken legislative action towards the mitigation of climate change, including, inter alia, regulations regarding the transportation sector and its emissions. For example, Norway was one of the world's first countries to introduce a $CO_2$ tax (1991), and despite not being a member state of the EU, it participates in the EU ETS and ESR [50,52]. Canada has legislated its own carbon pricing system [102], which is not yet implemented in every province, but only in seven of Canada's provinces. Since the Canadian carbon pricing scheme is formulated and regulated solely by the Canadian federal government, a change of governmental views on environmental issues may easily affect it. As for Norway, even though it is applied at the national level, in the transport sector the EU ETS only covers aviation, whereas the Norwegian $CO_2$ tax, set by the Norwegian Government and Parliament, affects aviation, road and sea transport (except domestic fisheries, which have a progressive refund scheme based on energy efficiency). Hence, Norway is also prone to a similar change in political will. That said, the Cabinet has signalled a stepwise increase in the $CO_2$ tax. Moreover, a number of other taxes and fees regulate transport in Norway, such as the road use tax, vehicle purchasing taxes and toll roads. Regarding the targeted legislation, electrification strategies are another critical institutional factor. Since the 1990s, Norway has been a pioneer regarding regulations towards EV penetration [57], which has led to a significant share of EVs in the Norwegian market for new cars in 2019, with a sales percentage of around 42% [56]. In comparison, Canada's EV legislation has been quite slow-paced, which has also translated into low sales, since newly registered electric cars amounted to just 3.5% of the overall car sales in the country [130]. However, the fact that both countries show limited progress in mitigating transport GHG emissions, as observed in Section 2, indicates that Norway's incentive policy is less effective than expected.

Compared to Norway, Canada demonstrates a wider adherence to fossil fuel combustion in passenger and freight transportation. A characteristic example is the Canadian railway system, which is mainly powered by diesel [136], whereas the Norwegian railway network is heavily dependent on electricity [87]. From an institutional perspective, this insistence is partly due to the Canadian federal government's hesitation to legislate for the use of alternative fuels while domestic oil and natural gas reserves remain abundant [134], as well to increased costs and other technological challenges. A potential institutional failure due to a contingent unwillingness to invest in infrastructure supporting green technologies is not as relevant in either country: both countries have already proposed significant investment plans towards the modernisation and environmental friendliness of their transportation systems [26,109].

Apart from these challenges, Canada is also prone to another potential institutional failure: the country is divided in provinces that have their own government and legislation. Local governments may not always comply with the federal government's legislation, thereby slowing down the transition of the transportation sector. A typical example of this malfunction is the government of Ontario's opposition to the nationwide legislation of the carbon pricing scheme, leading to a delay in its implementation. Such examples demonstrate that the independence of Canada's provincial governments may result in a slower transition or even a carbon lock-in.

*6.2. Interactions*

Knowledge diffusion is greatly affected by the interaction between the various agents and networks existing in an innovation system, since these interactions create a communication network between stakeholders. A typical example is the interaction between government policy and the enterprises operating on the examined system [12]. Interaction system failures are either caused by powerful dependencies that can heavily subdue innovative technologies, or by weak connections impeding innovation [152]. According to Carlsson and Jacobsson [151], these interactions leading to system failures are called strong and weak network interactions.

At first glance, both countries' systems consist of a vast variety of networks and actors, including transportation corporations, regulating authorities, governmental bodies and coordinating associations operating as umbrella networks. Nevertheless, a closer look provides a different perspective. Norway is characterised by the abundance of government-owned corporations, involved in almost every sub-sector of the transportation system [153]. On the other hand, Canada's transportation sector is dominated by private-owned entities, operating in every part of its innovation system [96]. The existence of a wide governmental control can lead to a better orchestration [154] and coordination [155] of transition policies, which is important for avoiding a technological lock-in. Nonetheless, the plurality of private stakeholders can also have a positive impact on the transition, since they strive for competitiveness and larger market shares [156], but also because the performance of private firms is usually dependent on their corporate social responsibility regarding environmental strategies [157]. It is also noteworthy that, in Canada, there are more stakeholders in each sub-sector, partly due to it being one of the largest and less homogenous countries in the world, with major regional, economic and systemic differences. Furthermore, in comparison with Norway, Canada plays an important role in the global transportation value chain, with many vehicle manufacturers operating in the country. These companies are not necessarily Canadian; many of them (especially in the automobile sub-sector) are headquartered in other countries. With the automotive industry acting on a global scale, the transition of the largely domestically-supplied Canadian transport sector will depend, to a certain extent, on international market trends rather than national initiatives. The same lesson applies to Norway, although Norway—in contrast—has no car industry other than supplying parts to car manufacturers abroad.

Another similarity of the countries examined is the fact that both are fossil fuel producers with a significant fossil fuel industry, in which local and international firms operate. Apart from a strong presence of national firms, Norway is also characterised by the operation of various European corporations, such as Shell and BP. The oil production landscape in Canada is similar, with the only difference being that Canadian enterprises have a smaller presence and the main stakeholders headquartered outside of Canada are mainly American firms, such as Chevron and Exxon Mobil. Although oil companies are not a component of the system analysed, their interactions with and within it are crucial, since the main energy source in the transportation sector are oil products such as gasoline and diesel, especially in the case of Canada and the oil-rich Western provinces. Their operation is deemed significant, since oil companies in both countries usually create strong lobbies in order to maintain their high revenues [158], which can lead to carbon lock-ins and significant interaction failures.

*6.3. Capabilities*

Another factor that may lead to a system failure is the inability of the system—and especially the unwillingness or inability of firms in the system examined—to adapt to new technological standards, in contrast with the local utilities in Norway, which took initiatives to develop a fast-charging infrastructure, as examined in Section 4. The transition to low-carbon transportation depends on the development of firms operating in this sector, since features such as financial resources, learning capacity and flexibility are required [12]. Furthermore, the transition of the transportation sector, in particular, requires the willingness of people to contribute to climate action, since a large share of energy is used for passenger transportation. This entails significant

behavioural changes. For instance, despite the provision of subsidies, many people avoid buying an electric vehicle, since electric vehicles remain costlier than conventional ones [159] and price is a significant aspect for citizens regarding fuel choice [160]. Considering that the price of gasoline in Canada is almost 50% lower than in Norway [161], Canadians are more hesitant regarding the purchase of an EV, since they still have access to relatively cheap fuel. Similarly, Norway's electricity prices are considerably lower than the EU average, making EVs even more economically attractive for Norwegians, since they can be cheaper to use, in addition to being cheaper to obtain due to the heavy subsidies. This preference towards cheaper fossil fuels is also reflected on Canada's automobile manufacturing industry. Although Canada is one of the major automobile manufacturers worldwide [89], it accounts for only 0.4% of the global electric car production [162]. A similar problem is also affecting the hydrogen vehicle market, since these vehicles have traditionally been more expensive than regular cars [163].

Excluding car transportation, the other sub-sectors of the system analysed are mainly driven by companies operating on each sub-sector. Regarding passenger transportation via rail, water or air, the choices for passengers are limited, although there is a recent tendency towards the electrification of ferries in Norway, as discussed in Section 4. The vast majority of passenger planes worldwide are fuelled by kerosene, with passengers not yet able to choose a considerably "greener" air transportation option. Nevertheless, they are able to choose transportation firms that demonstrate a more environment-friendly policy, if the costs remain at a similar level. On the other hand, companies operating on the industrial sector can demand that cargo transportation firms carry their products in "greener" ways: according to Touratier-Muller et al. [164], manufacturers are usually the ones forcing logistics companies to take environmental measures for the transportation of their products.

### 6.4. Infrastructure

According to Woolthius et al. [12], the unavailability of infrastructure is considered the last component of the system failure framework. Infrastructural failure is related to two different types of infrastructure [165]. The first category examines external factors related to communications and energy infrastructure. Norway and Canada are both two very developed countries with a high GDP per capita, and their communication networks are capable of supporting the operational requirements of the transportation sector. Their energy infrastructure is also modernised and capable of meeting the requirements of the sector. However, the availability of, and access to, fast chargers is still a potential bottleneck in both systems. Nevertheless, although it is considered an external infrastructural component, the energy grid has a greater importance regarding the transportation system, and especially the further diffusion of electric mobility. The transition towards a higher percentage of electric mobility contributes to the mitigation of climate change only if the electricity mix is characterised by high RES penetration. Norway's and Canada's electricity production is mainly based on hydropower; hence, the electrification of their transportation systems is significantly effective towards the reduction of GHG emissions. The second type of infrastructural challenges relates to internal factors, including scientific and technological infrastructure which can contribute to the diffusion of the examined technologies within the system. The main differences in the technologies used in Norway and Canada are related to road and rail transportation. Regarding road transportation, Norway is characterised by a very high (in comparison with other countries) percentage of battery electric cars, of about 10% [166]. On the other hand, battery electric cars in Canada amount only to 0.72% [130]. This contrast is also demonstrated in rail transportation. In Norway, 62% of rail tracks are electrified [78], whereas in Canada electric tracks are only used on urban modes of transportation such as metro train systems [136], partly due to the large nationwide distances and the respective capacities in the neighbouring United States. Concerning the other sub-sectors of the system, both countries exploit the same technologies. However, it is noteworthy that Norway has been experimenting on using electricity in other sub-sectors too, with some pilot projects related to marine and air transportation [86] showcasing the country's significant efforts to electrify not only road but also sea transport and short-distance domestic aviation.

Apart from the status quo of the existent technologies, infrastructural failure is also related to the potential for the further diffusion of these technologies and the penetration of new ones. Regarding electric mobility, Norway has two important advantages over Canada. As discussed above, electric mobility has already achieved an important penetration regarding road and rail transportation in comparison with Canada. The second advantage is the fact that Norway is almost 30 times smaller than Canada, and it is also more densely populated. Therefore, the expansion of electrification is considered easier in Norway, especially regarding charging stations and other equipment necessary for electric vehicles [149], even though the fast-charging network is dependent on initiatives from local utilities rather than governmental efforts. This advantage is also critical regarding marine, air and especially rail transportation. Because of its vast land area, Canada has hundreds of airports, seaports and an expansive railway network. Therefore, the purchase and installation of the required equipment for the transition to electric mobility is deemed far more expensive than in Norway. Electricity, though, is not the only alternative for fossil fuels. The mix of biofuels and the combustion of hydrogen and ammonia are three other solutions towards climate mitigation. Regarding the first two alternatives, Canada has demonstrated greater progress than Norway. Canada has a hydrogen association contributing to the diffusion of hydrogen-related knowledge among many stakeholders, including actors of the transportation network. Furthermore, it is characterised by great reserves of biofuels and is an important producer of biodiesel and ethanol [144]; the capacity in required reserves and a developed biofuels industry mean that a further penetration of biofuels can be achieved relatively easily in the country. In Norway, the production of biofuels has so far been scarce, despite a good access to potential raw materials, but there are plans to ramp up production significantly. Although the use of biofuels and hydrogen can contribute to the mitigation of climate change, they are also associated with important bottlenecks. Especially, the increased use of biomass has created several concerns regarding food security [167], as useful crops can be used for the production of biofuels instead of food supplies, while the accounting of biogenic carbon also remains controversial [168]. On the other hand, the diffusion of hydrogen combustion is not favoured in either country, since both are characterised by high hydropower penetration, instead of other renewables associated with green hydrogen, thereby significantly contributing to the penetration of electric mobility [149].

The key outputs of the system failures' comparative analysis are presented in Table 3.

**Table 3.** Key outputs from the comparative analysis based on the SF blocks.

| SF Blocks | Norway | Canada |
|---|---|---|
| Institutions | • Strong incentive policy has led to a high share of EVs, but GHG emissions remain high | • Policy inconsistencies due to provinces not always complying with federal legislation<br>• Hesitation to legislate against fossil fuels<br>• Slow-paced EV legislation |
| Interactions | • In both countries, fossil fuel companies create strong ties through lobbying processes, leading to carbon lock-in | |
| | | • The phenomenon is more intense in the oil-rich Western provinces |
| Capabilities | • Cheap electricity price provides an opportunity for the use of EVs, independent from other types of incentives | • Price hesitations for EVs compared to gasoline |
| Infrastructures | • Despite pledges, biofuel production has been scarce<br>• Spillover to green shipping indicates that innovation other than EVs can evade the examined failures<br>• The fast-charging network is dependent on initiatives from local utilities | • Electrification of rail very expensive due to the length of the network |

## 7. Results and Discussion

The abundance of fossil fuels and the economic prosperity of both countries have led to the continuously increasing activity of the transportation sector, accounting for approximately one quarter of GHG emissions in both countries. Institutionally speaking, both countries have taken action towards legislating various reforms in order to support the transition towards a "greener" transportation sector, including not only road, but also marine and air transportation. In the case of Norway there has been significant activity in the maritime sector, which is gearing up its domestic innovation capacities, with an emphasis on cutting-edge technologies in environmentally friendly shipping. While road EV diffusion is still the main innovation in the SIS of Norway, there are indications of a spillover from the electrification of road transport to sea transport, where Norwegian high-tech companies engage in R&D activities with the support of public investments. The lack of such activities in the case of Canada is evident in the legislative rigidities among the federal and provincial governments, as was the case with the dispute over the GHGPPA act and the support given to the domestic oil industry, which has strongly opposed the act. The interactions between the networks and actors of both innovation systems are generally stable, and in Norway actors are becoming increasingly integrated through joint initiatives. However, the level of emissions from transport have so far not decreased drastically, despite efforts in both countries, indicating the lack of a sincere effort by the system's actors to address these challenges. Both countries are characterised by an abundance of incumbents in every sub-network, but there seems to be great potential in the Norwegian maritime sector. Most incumbents are privately owned in Canada, while the picture is more mixed in Norway, with both publicly and privately owned companies, as well as public R&D schemes targeted at both the public and private sectors, with various implications—both positive and negative—for innovation coordination and the struggle for competitiveness. Another contingent barrier is the significant activity of oil companies in both countries, which may result in lock-ins, with oil industry accounting for an important portion of

both countries' exports. Excluding this challenge, the innovation systems of Canada and Norway are not considered highly susceptible to a system failure caused by the network agents' interactions.

Both countries' systems face challenges related to demand, some of which are also related to actors not directly involved in the system, such as the oil industry; hence, they are not so easily affected by changes within the systems. Regarding factors that are more closely related to system demand, Norway demonstrates a more robust transportation sector, since there is a larger demand for EVs and a landscape that is generally more supportive of the transition. This higher demand for EVs in Norway mainly results from the country's legislation, which has promoted the purchase of EVs with multiple financial stimuli, such as subsidies, lower toll rates, etc. The external infrastructural factors do not pose any significant challenges for neither country, since both Canada and Norway possess developed communication and energy networks. Regarding the internal factors, there are concerns that the fast-charger infrastructure is not developing rapidly enough, which could be a diffusion barrier in case demand rises beyond the capabilities of the existing infrastructure. Norway is facing fewer challenges on the sector of electric mobility, given the already existing network, even though it needs to be extended, but it is less capable of hydrogen or biofuel diffusion, although significant initiatives exist, especially regarding biofuels. In comparison, Canada has greater expertise and technical capacity, evident in the difficulties of Norway to increase the domestically produced share of biofuels despite the ambitious targets that were set. As a result, both countries may experience an infrastructural system failure regarding the penetration of electric vehicles and the transition of the transportation sector, either in the form of difficulties surrounding the charging infrastructure or limitations in the transition to alternative fuels.

In summary, Canada faces more risks in comparison with Norway, which may lead to a rockier road to decarbonisation. This obstacle stems mainly from Canada's demand, from institutional dynamics and, subsequently, from infrastructural requirements. Policy stringency is much more needed, therefore, in Canada, to counterbalance these challenges and counter the existent and potentially emerging bottlenecks, in order to overcome conventional technology lock-ins in the transportation sector. For this process, lessons learnt from Norway's progress can provide valuable policy insights for seizing opportunities and avoiding contingent threats. Norway's sustainable transportation strategy has been historically focused on incentive policies to boost the share of EVs. This led to an exponential increase in their shares, while also boosting public awareness. However, the limited progress achieved in mitigating transport GHG emissions suggests that even though such policies are useful, the evaluation of their impact is exaggerated, leading to a falsely cultivated image of transitional leadership, which even allows actors to ignore the lack of mitigation progress. Societal resistance to these policies led to a reduction of such incentives, in 2017, by 50%, without affecting the exponential growth trend, showing that these measures need to play a supplementary role as part of an effective decarbonisation strategy. In fact, cheap electricity prices from hydro-based power generation can provide a sufficient incentive for EVs due to the limited cost of use, an observation that can be useful for Canadian provinces like Quebec, with significant hydro-power potential. On the other hand, the spillover towards sustainable shipping, as well as smaller initiatives such as the growing interest in electric aviation, indicate that actors can be properly engaged in the process and actually plea for coherent strategies that place the emphasis on financing innovative technologies and infrastructure. Both Canada and Norway need to address these requests from their respective systems, adapting their strategies in harmonisation with actor initiatives, as in the case of the electric ferry and the development of a fast-charging network from local utilities in Norway. Channelling investments towards technological research and private-public sector collaboration can potentially limit the strength of regime actors, like the oil industry, as shown by the relative lack of extensive legislative rigidities in Norway that might threaten the transition.

## 8. Conclusions

The aim of this study is the analysis of the transportation sector in Norway and Canada from a systemic approach. This method contributes to the examination of the diffusion of innovative and sustainable technologies through the interaction between actors and networks. It also helps determine the barriers to their penetration into the studied systems. Our analysis does not have a single focus (e.g., electric vehicles), but investigates the broader spectrum of the transportation sector, including road, rail, marine and air transportation. The importance of actors, institutions, demand and knowledge was studied by means of the SIS framework, coupled with the SF framework, through the identification of potentially emerging or existing challenges in the adoption of climate-friendly, innovative technologies and practices.

The results indicate that Norway's EV incentive policy is less effective than expected, despite being considered pioneering. Low electricity prices from hydro-based power generation allow for the creation of sustainable strategies without relying heavily on additional financial motives. Instead, attention should also be given to the electrification of the other sectors, considering the high interest from the actors of the system. In the case of Canada, it was shown that regional differences have a significant impact on the development of a universal strategy. Multiple bottlenecks are introduced due to uncoordinated federal and provincial legislation; moreover, depending on the involvement in oil and natural gas production, fossil fuel powerhouses show different levels of influence. The same can be said of Norway, although public funding towards private-public collaborations tends to limit the strength of regime actors.

The systems examined in this research, with the application of the SIS and SF approaches, can be further analysed by using the multi-level perspective [169,170], innovation ecosystems [171], mapping frameworks [172] and risk scoping through the interaction of stakeholders [173,174]. This potential enhancement of the SIS approach would contribute to a broader examination of the systems studied, since the challenges regarding the diffusion of innovation would not be the sole focal point of the research; other aspects could also be investigated, such as landscape factors leading to significant changes in the transportation sector. Another prospect would be the use of the SIS to examine the interaction between the studied countries and international markets, so as to investigate the effect of global trading actors and networks on the diffusion of innovation in Norway and Canada, since transportation—and especially maritime and air transport—is closely related with international markets. The regional innovation system [175] can also be used in the Canadian transport system to formally study the regional differences outlined in this study. Finally, the Systems of Innovation frameworks can be applied to design transformative policy pathways [176,177], since their usage as climate policy support frameworks [178] and as part of integrative approaches [179] is becoming more common.

**Author Contributions:** Conceptualisation, A.N. and H.N.; methodology, H.N., K.K. and A.K.; validation, K.V., E.A.T.H. and A.N..; formal analysis, K.K., A.K., H.N. and A.N.; data curation, K.K., A.K. and E.A.T.H.; writing—original draft preparation, A.K., K.K., H.N., A.N. and H.D.; writing—review and editing, K.K., K.V., A.N., E.A.T.H. and H.D.; visualisation, K.K. and H.N.; supervision, A.N. and H.D.; project administration, H.D. All authors have read and agreed to the published version of the manuscript.

**Funding:** This research was funded by the European Commission Horizon 2020 Framework Programme, "PARIS REINFORCE" Research and Innovation Project, grant number 820846. The APC was funded by the National Technical University of Athens.

**Acknowledgements:** The most important part of this research is based on the H2020 European Commission Project "PARIS REINFORCE", under grant agreement no. 820846. The responsibility for the content of this paper lies solely with the authors. The paper does not necessarily reflect the opinion of the European Commission. The authors would like to thank Robbie Andrews from the CICERO Climate Economics Unit for the provision of some data on the Norwegian transportation sector.

**Conflicts of Interest:** The authors declare no conflicts of interest. The funders had no role in the design of the study; in the collection, analyses, or interpretation of data; in the writing of the manuscript, or in the decision to publish the results.

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
