# Peer review of "Many Miles to Paris: A Sectoral Innovation System Analysis of the Transport Sector in Norway and Canada in Light of the Paris Agreement"

_sustainability, doi:10.3390/su12145832_

Round 1
Reviewer 1 Report
1. The author needs to put page number
2. Revise Source in Fig 3 :
Source: [15Error! Bookmark not defined.], own elaboration
3. What is the objective (not the scope only) of this study? (Describe in Introduction).
4. According to this sentence in line 56, 57, 58 Please explain the strongly "answer" of what author describe in Introduction. The scope of this study is to map the transport sector of Norway and Canada from a low-carbon transition perspective". The answer of this were not in the concusion, nor in abstract. Please more focus in the conclusion and put it in anstract (shortened) so it will be easier for reader to understand the objective of this study.
56 The scope of this study is to map the transport sector of Norway and Canada from a low-carbon
57 transition perspective, by examining the structure and functions of their innovation systems and
58 assessing potential barriers that could hinder use and diffusion of sustainable technologies.
5. The writing were too plain. It is difficult to understand the main issue the author would like to tell the readers. Author need to revise to be more comprehensive.
6. Check the mistypes. Eq.
313 4.4 Knowledge, learing processes and technologies
Author Response
Reviewer 1
- The author needs to put page number
Response 1: The authors would like to sincerely thank the reviewer for their suggestions and productive comments. The format of the manuscript has been corrected to include page numbers according to the template. Initially, there was a problem with the section breaks, upon inserting landscape-oriented pages; we have now fixed this issue, by having pagination (on the top left of the template) resume across sections.
- Revise Source in Fig 3:
Source: [15Error! Bookmark not defined.], own elaboration
Response 2: All sources showing a bookmark definition error have been corrected. This was originally attributed to an error occurring upon saving the document as a PDF; wherever this remained problematic, we simply indicated the correct reference while removing the hyperlink.
- What is the objective (not the scope only) of this study? (Describe in Introduction).
Response 3: As suggested, the introduction has been revised to include a clear policy-oriented objective in line 60 of the introduction:
“Drawing from the examination and comparison of the systems, our objective is to gain insights into the effectiveness of implemented policies, inform on emerging pleas from the interior of the systems and provide lessons to policymakers towards the next steps of a transition to sustainable transport.”
- According to this sentence in line 56, 57, 58 Please explain the strongly "answer" of what author describe in Introduction. The scope of this study is to map the transport sector of Norway and Canada from a low-carbon transition perspective". The answer of this were not in the concusion, nor in abstract. Please more focus in the conclusion and put it in anstract (shortened) so it will be easier for reader to understand the objective of this study.
56 The scope of this study is to map the transport sector of Norway and Canada from a low-carbon
57 transition perspective, by examining the structure and functions of their innovation systems and
58 assessing potential barriers that could hinder use and diffusion of sustainable technologies.
Response 4: In accordance with comment 3, the scope has been supplemented with a clear objective:
“Drawing from the examination and comparison of the systems, our objective is to gain insights into the effectiveness of implemented policies, inform on emerging pleas from the interior of the systems and provide lessons to policymakers towards the next steps of a transition to sustainable transport.”
The conclusions in section 7 have been revised, with the insertion of a paragraph in line 903 that draws from the outputs of the analysis to clearly answer the objective described in the introduction:
“For this process, lessons learnt from Norway’s progress can provide valuable policy insights into seizing opportunities and avoiding contingent threats. The sustainable transportation strategy of Norway has been historically focused on incentive policies to boost the share of EVs. This led to an exponential increase in their shares, while also boosting public awareness. However, the limited progress achieved in mitigating transport GHG emissions hint that, even though such policies are useful, the evaluation of their impact is exaggerated leading to a falsely cultivated image of transitional leadership, which even allows actors to ignore the lack of mitigation progress. Societal resistance towards these policies led to a reduction of such incentives in 2017 by 50%, without affecting the exponential growth trend, showcasing that as part of an effective decarbonisation strategy these measures need to play a supplementary role. On the other hand, spillover towards sustainable shipping, as well as smaller initiatives, such as the growing interest in electric aviation, indicate that actors can be properly engaged in the process, and actually plea for coherent strategies that place emphasis on financing innovative technologies and infrastructure. Both Canada and Norway need to address these requests from their respective systems, adapting their strategies in harmonisation with actor initiatives, as is the case of the electric ferry and the development of a fast-charging network from local utilities in Norway. Channelling investments towards technological research and private-public sector collaboration can potentially limit the strength of regime actors, like oil industry, as evident from the relative lack of extensive legislative rigidities in Norway that threaten the transition.”
- The writing were too plain. It is difficult to understand the main issue the author would like to tell the readers. Author need to revise to be more comprehensive.
Response 5: Both the introduction and the conclusions sections have been heavily revised to include an objective that is answered from the outputs of this research to create a comprehensive narrative that leads to specific policy insights and recommendations (cf. comments on points 3-4). Also, the reasons behind selecting the two countries, in consideration of the sustainable transportation theme, has been better elaborated in lines 64-80:
“ The two countries were selected based on their profiles, which are similar in terms of GHG emissions, since the transportation sector has historically been a major emitter with similar shares in both systems, accounting for almost 24% of each country’s total emissions. Oil and gas extraction is a common dominant emitter [10], while the potential or penetration of renewables is also similar. The two countries are different in terms of innovation diffusion: although they both feature high shares of transport emissions as well as high contribution of renewable energy sources (RES) in the national energy mix [11], their EV shares in the transport mix differ significantly. This difference enables knowledge transfer between the two systems, leading to effective policy recommendations. Norway’s initiatives can provide valuable lessons to Canada. However, exchange of knowledge is not only flowing from Norway to Canada, since the progress achieved by Norway is showcased to be limited and does not place the country in a fundamentally better condition. From that perspective, Canada can also provide lessons, acting as a benchmark for Norway to properly evaluate policies, without exaggerating effectiveness. At the same time, both countries are locked into their domestic oil reserves but in a different manner: although Norway holds significantly less oil reserves, a share (usually less than 3 per cent) of the surplus of its state-owned oil fund (the world’s largest of its kind) is used to balance the national budget.
Finally, in line 86, the reasons behind selecting the two frameworks and an explanation of how they are intertwined with the objective of our study is better explained:
“In a review of sustainable transport research, Zhao et al. (2020) [14] showcased that the discussion surrounding transport policy places significant emphasis on sustainable transport indicators, which however failed to effectively influence policy. Instead, the two combination of the two qualitative frameworks selected in our study allows us to delve into market and public responses, which is a key consideration for assessing the effectiveness of policies without exaggerations [15].”
- Check the mistypes. Eq. 313 4.4 Knowledge, learing processes and technologies
Response 6: This, along with all other mistypes, have been corrected and very careful proofreading has been conducted by the authors, some of whom are also native English speakers.
The authors would again like to thank the reviewer for their helpful remarks and hope that the revised version of the manuscript lives up to the publication standards set by the Journal and the expectations of the reviewers.

Reviewer 2 Report
First, why do we need to compare Canada and Norway? The paper does not motivate this question correctly. Too many details are presented without any organization. To the very least authors should present some tables showing a comparison of studies.
Second, the discussion presented is nothing novel. The authors should provide more effort in figuring out how this review could help the countries, for example, in planning or other purposes.
Lastly, writing needs significant improvement. There are many proof-reading errors.
Author Response
Reviewer 2
1. First, why do we need to compare Canada and Norway? The paper does not motivate this question correctly. Too many details are presented without any organization. To the very least authors should present some tables showing a comparison of studies.
Response 1: The authors would like to thank the reviewer for their feedback. The selection of the countries has been elaborated in lines 64-80:
“The two countries were selected based on their profiles, which are similar in terms of GHG emissions, since the transportation sector has historically been a major emitter with similar shares in both systems, accounting for almost 24% of each country’s total emissions. Oil and gas extraction is a common dominant emitter [10], while the potential or penetration of renewables is also similar. The two countries are different in terms of innovation diffusion: although they both feature high shares of transport emissions as well as high contribution of renewable energy sources (RES) in the national energy mix [11], their EV shares in the transport mix differ significantly. This difference enables knowledge transfer between the two systems, leading to effective policy recommendations. Norway’s initiatives can provide valuable lessons to Canada. However, exchange of knowledge is not only flowing from Norway to Canada, since the progress achieved by Norway is showcased to be limited and does not place the country in a fundamentally better condition. From that perspective, Canada can also provide lessons, acting as a benchmark for Norway to properly evaluate policies, without exaggerating effectiveness. At the same time, both countries are locked into their domestic oil reserves but in a different manner: although Norway holds significantly less oil reserves, a share (usually less than 3 per cent) of the surplus of its state-owned oil fund (the world’s largest of its kind) is used to balance the national budget.”
In line 86, the selection of the frameworks and how they are intertwined with the objective is better explained, in order to be clearer to the reader why the presentation of these details follows the structure of the implemented frameworks:
“In a review of sustainable transport research, Zhao et al. (2020) [14] showcased that the discussion surrounding transport policy places significant emphasis on sustainable transport indicators, which however failed to effectively influence policy. Instead, the two combination of the two qualitative frameworks selected in our study allows us to delve into market and public responses, which is a key consideration for assessing the effectiveness of policies without exaggerations [15].”
2. Second, the discussion presented is nothing novel. The authors should provide more effort in figuring out how this review could help the countries, for example, in planning or other purposes.
Response 2: We genuinely believe the discussion is novel and has been backed by an impressive amount of literature and desk research. Please note that, in accordance with the reviewers’ suggestions, the introduction has been revised to include a clear policy-oriented objective in line 60 of the introduction:
“Drawing from the examination and comparison of the systems, our objective is to gain insights into the effectiveness of implemented policies, inform on emerging pleas from the interior of the systems and provide lessons to policymakers towards the next steps of a transition to sustainable transport.”
The conclusions in Section 7 have also been revised, with the insertion of a paragraph in line 903 that draws from the outputs of the analysis to clearly answer the objective described in the introduction:
“For this process, lessons learnt from Norway’s progress can provide valuable policy insights into seizing opportunities and avoiding contingent threats. The sustainable transportation strategy of Norway has been historically focused on incentive policies to boost the share of EVs. This led to an exponential increase in their shares, while also boosting public awareness. However, the limited progress achieved in mitigating transport GHG emissions hint that, even though such policies are useful, the evaluation of their impact is exaggerated leading to a falsely cultivated image of transitional leadership, which even allows actors to ignore the lack of mitigation progress. Societal resistance towards these policies led to a reduction of such incentives in 2017 by 50%, without affecting the exponential growth trend, showcasing that as part of an effective decarbonisation strategy these measures need to play a supplementary role. On the other hand, spillover towards sustainable shipping, as well as smaller initiatives, such as the growing interest in electric aviation, indicate that actors can be properly engaged in the process, and actually plea for coherent strategies that place emphasis on financing innovative technologies and infrastructure. Both Canada and Norway need to address these requests from their respective systems, adapting their strategies in harmonisation with actor initiatives, as is the case of the electric ferry and the development of a fast-charging network from local utilities in Norway. Channelling investments towards technological research and private-public sector collaboration can potentially limit the strength of regime actors, like oil industry, as evident from the relative lack of extensive legislative rigidities in Norway that threaten the transition.”
Along with the revisions explained in response 1, a clear framework is established that builds a comprehensive narrative leading to specific policy insights and recommendations.
3. Lastly, writing needs significant improvement. There are many proof-reading errors.
Response 3: The authoring team are collectively highly experienced researchers with a longstanding publication record in high-impact journals in the field; part of the team are also native English speakers, and the manuscript has been thoroughly reviewed by the team. As evident from the tracked changes, some modifications have been made to the manuscript towards improving the quality and level of the English language used, while pagination issues and errors emerging from Word-to-PDF conversion have also been corrected.
The authors hope that the revised version of the manuscript lives up to the publication standards set by the Journal and the expectations of the reviewers.

Round 2
Reviewer 1 Report
Thank you for the revisions. But there are some other major revision you have to do, such as :
- Abstract should describe the whole research process and conclusion data. It is not only talk about what yo are going to do on the research. Your abstract did not show the overall result of this research.
- Separate the Conclusion and Result & Discussion, so that will be easier for reader to understand the conclusion.
- Please more focus on the objectives. Can skip the unnecessary words that only to make the manuscript longer, and not understandable.
Author Response
1. Thank you for the revisions. But there are some other major revision you have to do, such as:
Abstract should describe the whole research process and conclusion data. It is not only talk about what yo are going to do on the research. Your abstract did not show the overall result of this research.
Response 1: The authors would like to sincerely thank the reviewer for their suggestions and productive comments. Abstract has been revised to clearly state key results of the study.
2. Separate the Conclusion and Result & Discussion, so that will be easier for reader to understand the conclusion.
Response 2: According to the reviewer’s suggestion, the final section was split into “Results and Discussion” and “Conclusions”. Apart from the discussion on the results of the study, a paragraph has been inserted (and highlighted) in the conclusions to clearly summarise the key conclusions.
3. Please more focus on the objectives. Can skip the unnecessary words that only to make the manuscript longer, and not understandable.
Response 3: To make the study more understandable and assist the reader in navigating through the plethora of information presented, exhibits have been added to summarise key intermediate outputs from all the steps of the analysis creating a more comprehensive narrative. The addition of Figure 6 aims to present the methodological approach more clearly. Tables 1 & 2 present key outputs from the two SIS that drive the discussion in the following segments. Table 3 also presents the key outputs from the SF analysis that lead directly to the key results. Skipping information might imply the lack of activity on a certain block, which could have been misleading if not properly discussed. We believe that through these changes it is easier for the reader to understand the methodological process, the takeaways from each step, and how they build up to the main conclusions of the study.
The authors would again like to thank the reviewer for their helpful remarks and hope that the revised version of the manuscript lives up to the publication standards set by the Journal and the expectations of the reviewers.

Reviewer 2 Report
The authors responded to the comments provided by the reviewer. However, the reviewer still has some concerns. The reviewer believes it would be better for the readers to understand the information synthesized if a few exhibits are provided. For example, the authors could present important characteristics of Norwegian and Canadian transport systems in an exhibit instead of listing them one by one in Sections 4 and 5. There are other areas where similar work can be done.
Minor:
"(2020)" should be removed from the texts in line 87.
Author Response
The authors responded to the comments provided by the reviewer. However, the reviewer still has some concerns. The reviewer believes it would be better for the readers to understand the information synthesized if a few exhibits are provided. For example, the authors could present important characteristics of Norwegian and Canadian transport systems in an exhibit instead of listing them one by one in Sections 4 and 5. There are other areas where similar work can be done.
Response 1: The authors would like to thank the reviewer for their feedback. To make the study more understandable and assist the reader in navigating through the plethora of information presented, exhibits have been added to summarise key intermediate outputs from all the steps of the analysis creating a more comprehensive narrative. The addition of Figure 6 aims to present the methodological approach more clearly. Tables 1 & 2 present key outputs from the two SIS that drive the discussion in the following segments. Table 3 also presents the key outputs from the SF analysis that lead directly to the key results. Skipping information might imply the lack of activity on a certain block, which could have been misleading if not properly discussed.
Accordingly, the final section was also split into “Results and Discussion” and “Conclusions”. Apart from the discussion on the results of the study, a paragraph has been inserted (and highlighted) in the conclusions to clearly summarise the key conclusions.
We believe that through these changes it is easier for the reader to understand the methodological process, the takeaways from each step, and how they build up to the main conclusions of the study.
"(2020)" should be removed from the texts in line 87.
Response 2: Revised according to the reviewer’s comment.
The authors hope that the revised version of the manuscript lives up to the publication standards set by the Journal and the expectations of the reviewers.

Round 3
Reviewer 2 Report
Please provide Fig. 6 in a higher resolution.
Author Response
Please provide Fig. 6 in a higher resolution.
Response: The authors would once more like to thank the reviewer for their devotion and comments. Figure 6 has been restructured to enlarge fonts and submitted as a source file to MDPI.